# Modular pipeline for reconstruction and localization of implanted intracranial ECoG and sEEG electrodes

**Daniel J. Soper** [1,2]*, **Dustine Reich**[1,3], **Alex Ross**[4], **Pariya Salami**[1,2], **Sydney S. Cash**[1,2], **Ishita Basu**[4‡], **Noam Peled**[5,6‡], **Angelique C. Paulk**[1,2‡]

**1** Department of Neurology, Center for Neurotechnology and Neurorecovery, Massachusetts General Hospital, Boston, MA, United States of America, **2** Department of Neurology, Harvard Medical School, Boston, MA, United States of America, **3** Department of Neurology, Brigham and Women's Hospital, Harvard Medical School, Boston, MA, United States of America, **4** Department of Neurosurgery, University of Cincinnati College of Medicine, Cincinnati, OH, United States of America, **5** Department of Radiology, Athinoula A. Martinos Center for Biomedical Imaging, Massachusetts General Hospital, Charlestown, MA, United States of America, **6** Harvard Medical School, Boston, MA, United States of America

☯ These authors contributed equally to this work.
‡ IB, NP, and ACP are Joint Senior Authors
* djsoper@mgh.harvard.edu

**Data Availability Statement:** All relevant data for this study are publicly available in Data Archive BRAIN Initiative repository (https://doi.org/10.18120/gpgp-4r37).

## Abstract

Implantation of electrodes in the brain has been used as a clinical tool for decades to stimulate and record brain activity. As this method increasingly becomes the standard of care for several disorders and diseases, there is a growing need to quickly and accurately localize the electrodes once they are placed within the brain. We share here a protocol pipeline for localizing electrodes implanted in the brain, which we have applied to more than 260 patients, that is accessible to multiple skill levels and modular in execution. This pipeline uses multiple software packages to prioritize flexibility by permitting multiple different parallel outputs while minimizing the number of steps for each output. These outputs include co-registered imaging, electrode coordinates, 2D and 3D visualizations of the implants, automatic surface and volumetric localizations of the brain regions per electrode, and anonymization and data sharing tools. We demonstrate here some of the pipeline's visualizations and automatic localization algorithms which we have applied to determine appropriate stimulation targets, to conduct seizure dynamics analysis, and to localize neural activity from cognitive tasks in previous studies. Further, the output facilitates the extraction of information such as the probability of grey matter intersection or the nearest anatomic structure per electrode contact across all data sets that go through the pipeline. We expect that this pipeline will be a useful framework for researchers and clinicians alike to localize implanted electrodes in the human brain.

## Introduction

Patients with medication-resistant epilepsy can be candidates for surgical treatments for alleviating seizure burden that typically involve intracranial electrode implants to accurately identify

**Funding:** The authors report the following sources of funding: Support included Tiny Blue Dot foundation, https://www.tinybluedotfoundation.org/, to SSC, DJS, DR, and ACP. SSC was funded by NIH grants NINDS R01- NS062092, 1K24NS088568, R01-NS079533, R01-NS072023, and Massachusetts General Hospital Executive Committee on Research (MGH-ECOR). Some of this research was sponsored by the U.S. Army Research Office and Defense Advanced Research Projects Agency (DARPA), https://www.darpa.mil/, under Cooperative Agreement Number W911NF-14-2-0045 issued by ARO contracting office in support of DARPA's SUBNETS Program. PS is supported by DoD (CDMRP FY21 Epilepsy Research Program W81XWH-22-1-0315). IB and AR were partially funded by the NIMH grant 1 R21 MH127009-01A1. The views and conclusions contained in this document are those of the authors and do not represent the official policies, either expressed or implied, of the funding sources. The funders had and will not have a role in study design, data collection and analysis, decision to publish, or preparation of the manuscript.

**Competing interests:** The authors have declared that no competing interests exist.

**Abbreviations:** Pre-op, pre-operative (before intracranial electrode implant; Post-op, post-operative (after intracranial electrode implant; DICOM, Digital Imaging and Communications in Medicine; MPRAGE, magnetization-prepared 180 degrees radio-frequency pulses and rapid gradient-echo; Ax, Axial; NifTI, Neuroimaging Informatics Technology Initiative; MRI, Magnetic Resonance Imaging; CT, Computed Tomography; RAS, Right Anterior Superior; OS, Operating System (Windows, Mac, Linux; MMVT, Multi-Modal Visualization Tool; ECoG, electrocorticography; sEEG, stereoelectroencephalography; iEEG, intracranial electroencephalography; Reconstruction, recreation of a brain's surfaces and/or volumes from an MRI file.

the seizure onset area(s) [1–6]. These surgically implanted electrodes consist of two types: 1) contacts along thin tubes that extend into the brain (also intraparenchymal depths) targeting subcortical brain regions or deep cortical structures, which we call stereotactic electroencephalography (sEEG) electrodes, or 2) grids and strips on silastic sheets that lay on the cortical surface, which we call electrocorticography (ECoG) electrodes. These electrodes are designed to record neural activity, intracranial EEG (iEEG), with the goal of delineating the seizure onset zone (SOZ) for further clinical interventions such as resection, ablation, or stimulator placement [1,2,7–10]. Intracranial brain recordings via these electrodes also allow for the study of brain activity during other cognitive and physiological processes otherwise not possible with non-invasive measures [1,11–14]. In addition, multiple groups have begun using sEEG and ECoG electrodes in clinical trials to identify locations for deep brain stimulation to treat severe depression [15–18].

A key requirement to achieving clinical and research goals is the precise localization of the implanted electrodes relative to brain regions. Most methods for localization use spatial co-registration of the pre-implantation MRI with a post-implantation CT or MRI to produce a visual representation of the implanted electrodes [5,19–26]. Several implementations involve all-in-one software packages that provide validated pipelines for localizing electrode locations [1,19,22–30], and some pipelines require an experienced radiologist to identify the brain locations of each electrode contact manually [1,5]. In addition, some co-registration pipelines require the pre-operative and post-operative scans to first be transformed into a standardized brain space (e.g., Montreal Neurological Institute (MNI) space) before identifying electrode locations [28,29]. There are various methods for a user to achieve their goal when attempting to localize implanted electrodes, but here we lay out a protocol involving multiple software packages that we have used in a large data set (>260 patients) and which we have used to address many neuroscientific and clinically-relevant goals [5,6,11–14,30–38].

Our protocol incorporates several different software packages and methods and facilitates the incorporation of additional packages and their outputs depending on user skill and need. The overarching outputs of the protocol are (1) visualization of the electrodes, (2) localizations of the electrodes relative to the brain, (3) algorithmic localization of electrodes to brain structures, (4) data curation into an anonymized, shareable format, and (5) additional quantitative tools for identifying specific electrode placement relative to white matter, grey matter, subcortical regions, or whether contacts are outside brain. Regardless of the use case, we attempt to lay out the pipeline to be clear and easy to follow to maximize reproducibility across users.

We developed and designed our protocol pipeline to achieve three major benefits: first, (1) it is modular in execution with flexible optional outputs; (2) it is easy to use with steps explained in plain terms, even for naïve users; and (3) it maps electrodes to the patient's native space for electrode localization.

First, modularity both in the steps and the output is of particular importance so that we have a multi-purpose pipeline which does not require knowledge or execution of all steps in the entire protocol. For example, we anticipate that clinicians may only require the 2D or 3D visualization outputs from this pipeline, while neuroscience researchers may want to quantify electrode locations with respect to brain regions and anatomical features. To this end, we note what steps are optional for specific final products. Another aspect of this modularity is that other pipelines or changes to the programming code can be added by the user with relative ease. All of the code for this protocol is documented and open to changes as desired, and we include references to alternative software and methods for different steps within the protocol.

Second, our protocol is built to be accessible to users with minimal programming experience, allowing even naïve users to follow the steps to achieve the output of interest. The relevant MATLAB code uses mostly basic MATLAB functions with a few additional provided

functions to assist users who do not have access to paid MATLAB toolboxes. The Python and Bash commands are also easily implemented and rarely need to be altered. Orienting radiological images (MRI) in 3D space, creating images, and saving images are all automatic. This is particularly advantageous in the sense that a potential user need not have specialized knowledge of neuroanatomical landmarks to be able to perform successful coregistration of the pre-operative MRI (before electrode implants) and post-operative images (CT or MRI with implanted electrodes) [25,28]. We also include information on how to install the necessary software packages, as well as information which will help naïve users to follow this pipeline with minimal effort, including videos and example data [39].

Finally, our images and electrode coordinates remain in the patient's native space. This is especially important for patients with substantial abnormalities in their brain structure due to lesions or prior surgery. Such alterations in anatomy can make morphing into a common brain space problematic. Relevant to this point, we further include two different approaches to map electrodes to brain regions identified through the automatic parcellation and segmentation of the patient's brain into common atlas space [5,30,40–42]. We incorporate the preservation of native space and the transformation in a common space in order to improve the flexibility of this pipeline.

Notably, our pipeline is not a single software package but is a methodology that takes advantage of multiple software packages, utilizing the strengths of each package for an expandable and flexible pipeline for electrode localization (**S1 Fig**). With the minimum installation of 2 pieces of software, and a total of 5 downloads, we demonstrate a working pipeline which can create 2D and 3D visualizations of iEEG electrodes along with automatic, algorithmic localization of electrode contacts to brain structures and brain regions using multiple approaches. This protocol has been followed by multiple users, ranging from individuals new to programming to adept programmers, to reconstruct electrode locations in over 260 patients with iEEG across multiple hospitals. The pipeline also incorporates software for de-identified output using the iEEG BIDS formatting which is crucial in accelerating data sharing [43]. In addition, in the hands of advanced users, portions of the software in the pipeline identifies electrode contact location relative to brain features and permits visualization of real-time activity on 3D brain surfaces [30].

## Materials and methods

The protocol and pipeline pertains to the mapping of electrode locations for recorded intracranial neural activity from patients undergoing invasive monitoring, typically for medication-resistant epilepsy [39]. Participants undergoing invasive monitoring could be implanted with stereo-electroencephalography (sEEG) and/or ECoG grids and strips to locate epileptogenic tissue in relation to essential cortex. In the examples used here, depth electrodes (Ad-tech Medical, Racine WI, USA, or PMT, Chanhassen, MN, USA) with diameters 0.86–1.27 mm and 4–16 platinum/iridium-contacts 1–2.4 mm long with inter-contact spacing ranging from 4–10 mm (median 5 mm) were placed based on the clinical indications for seizure localization determined by a multidisciplinary clinical team independent of this research.

The series of steps for this pipeline are briefly listed below with the relevant references (**Fig 1**). However, the detailed steps (including video) of the protocol described here is published on protocols.io, https://www.protocols.io/view/modular-reconstruction-and-co-registration-of-imag-5qpvornedv4o/v2 [39]) and is included for printing as **S1 File** with this article. The protocol is the definitive guide for installation and execution of the pipeline we describe here.

## Step 1: Retrieve Images

Post-operative and Pre-operative imaging accessed to be used in the reconstruction

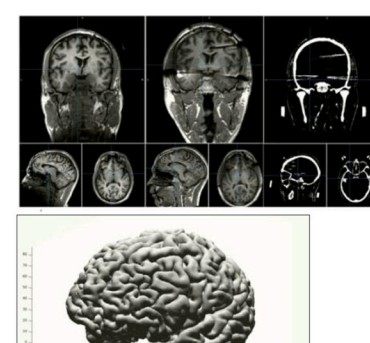

## Step 2: FreeSurfer Output Detailing Brain Regions

Creates the 3D surface files of the brain from the pre-op imaging

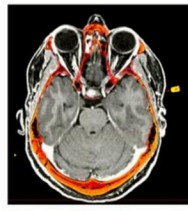

## Step 3: Imaging Co-Registration with FreeView

Align post-op imaging with pre-op imaging

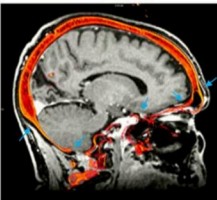
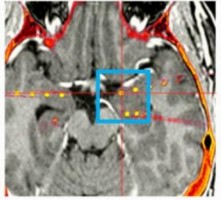

## Step 4: Identifying electrode locations in RAS coordinate systems

Use Freeview to select electrode contact coordinates in 3D space

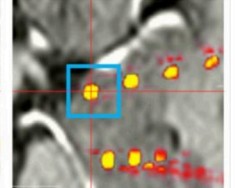

## Step 5: Snapping the Electrode Grid

For grid and strip (surface) electrodes only, project grid coordinates onto the pial surface

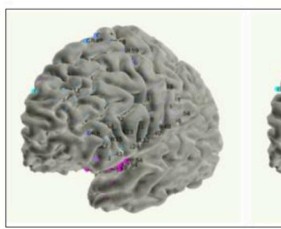
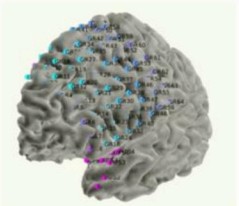

## Step 6: Creating Images of the MRI with the Electrode Overlay

Use MATLAB code to create images displaying electrode location in pre-operative MRI imaging

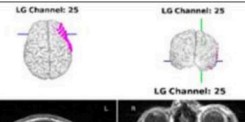
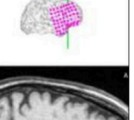

## Step 7-10: Electrode Localization, Data Sharing, and Visualization

Algorithmically localize electrodes to brain regions following parcellations with selected brain atlas. Also includes data anonymizing and sharing steps, as well as extra tools for visualizing and determining other anatomical landmarks like grey matter boundaries.

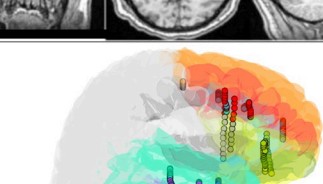
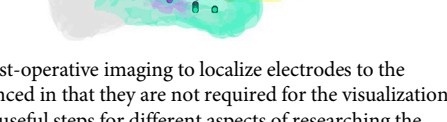

**Fig 1. Protocol pipeline.** The steps from the original pre-operative and post-operative imaging to localize electrodes to the resultant visualizations for a number of different uses. Steps 7–10 are advanced in that they are not required for the visualization of the image location relative to the pre-operative MRI. However, they are useful steps for different aspects of researching the human brain.

In terms of the necessary computational hardware to support this pipeline, the reconstruction steps using FreeSurfer are the most demanding. Therefore, we recommend that the system meet their requirements (https://surfer.nmr.mgh.harvard.edu/fswiki/rel7downloads) as the rest of the pipeline is less computationally intense. There are four main hardware requirements: (1) Intel processor supporting AVX instructions, (2) 8GB of RAM for the reconstruction, or 16GB of RAM for better graphical viewing, (3) a 3D graphics card with its own memory and accelerated OpenGL drivers, and finally (4) around 300MB of free hard drive space per processed subject.

Though this protocol was created with first-time, naïve users in mind, it must be understood that this protocol is intended as less of a pedagogical tool and more as a guide for researchers familiar with the input (e.g. brain imaging), output, and motivations of localizing electrodes in the brain. As such, some of the steps described below may take much longer for users new to these concepts. For this reason, in the online protocol, we include videos which capture the sequence of steps needed such that users have both written and visual instructions to follow the steps [39]. Further, the protocol is designed to have multiple, parallel outputs such that users can stop at one step and not proceed to the next steps (**Figs 1 and S1; Table 1**).

### Step 1: Retrieve images, 5–10 minutes

Following implant, the pre-operative T1-weighted MRI before contrast is acquired along with the post-operative CT or MRI with electrodes implanted. These images do not have to be anonymized or named anything in particular at this step.

### Step 2: Freesurfer reconstruction output, 3–10 Hours

To reconstruct the brain regions and parcellate/segment the brain regions from the MRI, we used the recon-all function from the FreeSurfer scripts (http://surfer.nmr.mgh.harvard.edu) on the pre-operative T1-weighted MRI [41,42,44–61] followed by functions which subdivided brain regions based on the Desikan-Killiany-Tourville Cortical Atlas (DKT) using default parameters [42]. This step is time-consuming and is not necessary if only the 2D visualizations from Step 6 are desired. However, there are different paths for different outputs associated with this pipeline which are detailed both in **Table 1** and in two flow charts (**Figs 1 and S1**). For the output from the "Basic" or "Advanced" pipelines, we find starting Step 2 as soon as possible, even before the initial electrode implant, can offset the amount of time the FreeSurfer pipeline takes.

**Table 1. Required and optional steps for arriving at specific output in the protocol.**

| | Basic Pipeline | 2D Visuals Only | Advanced Pipeline |
|---|---|---|---|
| **Protocol Steps** | Steps 1–6 | Steps 1 and 3–6 | All Steps, 1–10 |
| **Output** | • RAS coordinates<br>• Reconstructed Surface<br>• 2D Visuals of electrodes on the preoperative MRI<br>• 3D Visuals of electrodes in the Freesurfer reconstructed brain | • RAS coordinates<br>• 2D Visuals of electrodes on the preoperative MRI | • RAS coordinates<br>• Reconstructed Surface<br>• 2D Visuals<br>• 3D Visuals<br>• ELA probabilities<br>• BIDS Files Structure<br>• EVL localizations<br>• Nearest Anatomical Structure<br>• Grey/White Distances<br>• De-faced Imaging |

This table lists the three main use cases for this pipeline at the top with their corresponding steps and output below.

## Step 3: Imaging co-registration with freeview, 5 minutes

The pre-operative T1-weighted MRI is co-registered with a post-operative CT using volumetric image co-registration procedures using FreeSurfer tools (http://surfer.nmr.mgh.harvard.edu), [41,42,44–61] specifically using Freeview and default parameters. These procedures rely on linear transformations of the post-operative imaging onto the pre-operative imaging. A key point is that all the imaging is registered to a single identified pre-operative T1-weighted MRI scan. As such, if multiple CT scans are performed, each can be registered to the single pre-operative T1-weighted MRI scan sequentially such that the coordinates can be mapped to the same space. A short video of how this is performed is included in the protocols.io publication [39]. Optionally, we have also found we could use Mango [62–64] or LeadDBS [29] for this alignment step, since the coordinates just need to be in the native space of the pre-operative MRI (**S1 Fig**).

## Step 4: Identifying RAS coordinates, 0.5–2 hours

Electrode coordinates are manually determined from the overlaid electrodes (from the post-operative CT or MRI) in the patient's native space [5]. The length of time this step takes depends on many factors including: the resolution of the post-operative scans, the number of electrodes, knowledge of the relative locations for each electrode, and the skill level of the user. There is a short video example of identifying these coordinates in the protocols.io publication [39].

## Step 5: Snapping the electrode grid, 30 minutes

When grid and strip electrodes on the surface of the brain are implanted, there is a shift and compression in the brain as a result of the surgery, which can lead to the electrodes being incorrectly mapped relative to the pre-operative MRI. Various researchers have identified algorithms to shift the electrodes to the surface of the brain while constraining the relationships between contacts, though these details are reported and validated elsewhere [21,65,66]. We use the extracted and smoothed surface of the pre-operative MRI as a target for snapping the grids and strips, specifically using the method detailed in Dykstra et al. [5]. This method was selected for its ease of use with the pipeline, but many other pipelines can handle this step as well if desired (**Table 2, S1 Fig**) [5,21,65,66].

**Table 2. Features of this pipeline and alternatives.**

| Pipeline Name | Image Registration | Freesurfer-Compatible | Snap to Cortical Surface | Automatic Visualizations | Electrode Region Labelling | Native Space | Neural Data Processing | Anonymization |
|---|---|---|---|---|---|---|---|---|
| **This Pipeline** | Manual | Yes | Dykstra et al. [5] | 2D, 3D, Length-wise | Surface & Volume-Based | Yes | No | BIDS |
| **Lead DBS** | Automatic | Yes | Yes | 2D, 3D | Subcortical | Possible | No | No |
| **Brainstorm** | Automatic | Yes | Yes | 2D, 3D | Surface-Based | No? | Yes | No |
| **iELVis** | Automatic/Manual | Yes | Dykstra et al. or Yang et al. [5,21,66] | 2D, 3D | Surface-Based | Yes? | No | No |
| **LeGUI** | Automatic | No | Hermes et al. [65] or Normal Vector Projection | No | Volume-Based | Yes | No | No |
| **Stolk/Fieldtrip** | Automatic | Yes | Dykstra et al. [5,25] | No | Surface & Volume-Based | Yes | Yes | No |

This table lists eight aspects of an electrode localization pipeline with six pipelines along the left side, including this pipeline.

## Step 6: Creating images of the MRI with the electrode overlay, 30 minutes

To communicate the electrode locations relative to the MRI among the interested teams, we generate a.pdf (Adobe Acrobat) document (**S2 Fig**) which includes per-page, individual contact locations relative to the 3D brain and the MRI axes (coronal, sagittal, and horizontal sections). Further, we visualize the length of the depth electrode in a re-sliced coronal view along the long axis of the depth electrode. This step can be run with only 2D output (the electrode overlaid onto the MRI slice) as well, which does not require 3D surfaces (which are produced from FreeSurfer). This step uses a custom MATLAB script that we developed and which is included in the associated GitHub page (https://github.com/Center-For-Neurotechnology/Reconstruction-coreg-pipeline) and detailed in the online protocol [39].

## Step 7: Multi-Modality Visualization Tool-lite (MMVT-lite), 10 minutes

The Multi-Modality Visualization Tool (MMVT) is an open-source python package developed by NP that converts the FreeSurfer reconstruction output into a Blender (blender.org) [67] file for three-dimensional visualization of electrode locations which can incorporate MRI, Freesurfer output, RAS mapping, and even evoked potentials into the visualization (https://mmvt.mgh.harvard.edu/) [30]. The original MMVT output has been highlighted in a number of publications and journal covers in the past few years [17,31,68,69]; however, the full MMVT installation requires a considerable number of dependencies for installation and has not been updated to use the latest version of blender.

As such, for the purposes of our protocol, we introduce MMVT-lite (https://github.com/pelednoam/mmvt_lite), which incorporates FreeSurfer commands that include parcellating the brain, mapping to different brain atlases, and the electrode labelling algorithm (ELA, (https://github.com/pelednoam/ieil)) [40] but does not output a blender file. MMVT-lite requires fewer dependencies than the original MMVT version, can be installed on any Mac or Linux machine (or Windows virtual machine) and uses ELA for automatic labelling of electrodes relative to different brain regions. NP developed ELA to identify the nearest brain region label by identifying the probability of overlap of an expanding area (cylinder) around each electrode with identified brain structure labels corresponding to grey matter (whether cortical or subcortical) using purely anatomical approaches. We map electrodes to regions in a given location which can be flexibly chosen within FreeSurfer, using the DKT 40 atlas in combination with a subcortical mapping [41,42,52,61]. As the ELA method involves arriving at probabilities that the contacts or the bipolar pair of electrodes are in that labelled brain region based on proximity to labelled nearby surfaces of the brain structures, a single contact can have multiple probabilities above zero if the contact is near different areas. This data is output in the form of a spreadsheet (.xls or.csv) with every brain region given a probability that each electrode contact is located in that region. Notably, parcellating the brain, which involves mapping brain structures onto the brain using common brain atlases, can also be independently done of MMVT-lite using FreeSurfer or SPM tools [41,42,46,60,70].

## Step 8: iEEG BIDS formatting of channel labels and file formats, 30–40 minutes

Another feature we have incorporated into our pipeline involves converting the neural and electrode labelling into the intracranial electroencephalography Brain Imaging Data Structure (iEEG BIDS) format and folder structure including listing the electrodes based on iEEG BIDS formats [43]. As such, we included our modified code from the Bids Starter Kit code (https://bids-standard.github.io/bids-starter-kit/) which includes reformatting the neural, imaging,

and electrode localization data into the Brain Vision format with the BIDS folder structure (https://bids-standard.github.io/bids-starter-kit/tutorials/ieeg.html).

### Step 9: Extra measurements of electrode location relative to brain features, 20–30 minutes

The next step involves running custom MATLAB code developed by ACP that uses the Free-Surfer output, RAS file, and MMVT file output structure to perform different calculations regarding electrode location, label electrodes based on DKT or other selected brain atlas maps and produce output that matches the iEEG BIDS formatting. This also includes using a second approach for labelling each electrode based on the brain region location.

A second, novel, optional type of labelling that we developed for this pipeline for further electrode location validation, labelled Electrode Volume Labelling (EVL), involves mapping electrode locations relative to brain volumes circumscribed by the parcellated brain region maps. To do this, we export the FreeSurfer volume parcellations into 3D files (.stl files) using 3DSlicer (https://www.slicer.org/). The steps involve importing files from the FreeSurfer output folder and then saving the volumes per brain label: 1) load the brain.mgz and aparc.DKTatlas+aseg.mgz files from the SurferOutput folder; 2) save the output volumes of the parcellations: save (export) the.stl file to the selected folder. The resultant saved volumes are imported into MATLAB (MATLAB 2020b). Then the MATLAB function alphaShape is used to generate enclosing volumes per brain region label. The second MATLAB function, inshape, is then used to determine if an electrode was within the brain region volume (such as a cortical ribbon for a cortical label). Notably, for this step, we check that the T1 used in the original co-registration (Step 3) is in the same orientation and location as the FreeSurfer output.

For identification of electrode location relative to grey and white matter, we measure the orthogonal Euclidean distance from the center of each bipolar pair of electrodes to the nearest reconstructed vertex of the pial and white matter surfaces generated from FreeSurfer tools following colocalization [41,42,44–61]. Inter-contact distances between each contact is re-calculated using Euclidean measures. We classify electrode sites as in the grey matter, subcortical regions, white matter, and pial surface by identifying the colocalized location if the location is within the grey matter volume, the white matter volume, and the reconstructed subcortical volumes in the participants' native space. The classification of the sites relative to the surfaces is done using the MATLAB inpolyhedron function [14,71]. In addition, we use the MATLAB functions alphaShape and inshape to identify electrodes on the pial surface but not within the cortex.

### Step 10: Defacing the MRIs and CTs for deidentified data sharing, 10–15 minutes

For sharing data following FAIR data practices [72], it is essential that the data be de-identified to protect the identity of the patients. A number of algorithms can be used to de-face the scans [1,5,19,27,48,49,51,73]. However, as there can be errors in removing facial features from scans while allowing the scans to be useable for future reconstructions [74], we use the fieldtrip de-facer function for manual defacing of each scan [25,27].

### Ethics statement

All imaged patients voluntarily participated after fully informed consent as monitored by the Partners Institutional Review Board covering Brigham and Women's Hospital (BWH) and Massachusetts General Hospital (MGH). The full consent process is both verbal and written in that

the initiation of consent starts with a conversation and the final consent is given through a written form. Participants were informed that participation would not alter their clinical treatment in any way, and that they could withdraw at any time without jeopardizing their clinical care.

## Expected results

There are multiple outputs from this pipeline which can help answer multiple clinical and research questions (**Fig 1**, **Table 1**). We can find the location of the electrodes in a patient's native brain space, thus allowing us to visualize the electrodes in the patient's brain to be shared as.pdf documents in an easy-to-understand format (**Figs 1**, **2 and S1**; **Table 1**). We also can use one of many brain atlases, which can be used to parcellate the brain into identified brain regions in FreeSurfer [41,42,44–61] followed by the electrode labelling algorithm (ELA; [40]) or the Electrode Volume Labelling (EVL). This allows us to algorithmically determine the locations of electrode contacts in the brain. This brain region localization alone has been useful for following seizure activity, planning stimulation, and localizing neural activity relative to cognitive processes [5,6,11–14,30–38]. Finally, this output data is easily converted into a universal format so that we can share our data with the greater scientific community (iEEG BIDS; [43]).

A primary design feature and requirement of this pipeline is to generate a shareable visual representation of the electrodes in the brain in a format which would be familiar to clinicians with a clear layout and at least three main views of the pre-operative MRI (coronal, sagittal, and horizontal; **Figs 2B**, **S1 and S2**; **Table 1**). The visualization is used to validate whether electrodes are implanted in the desired brain regions and to map neural epileptiform activity and the seizure onset zone (SOZ) to specified brain regions. This pipeline has been shared with clinical teams at Massachusetts General Hospital and Brigham and Women's Hospital for nearly 15 years, which accounts for over 260 patients as of publication [5,6,11–14,30–38]. The end product is a shareable.pdf file (Adobe Acrobat; **S1 File**) that provides clinical and research teams with the appropriate electrode locations and labels, the three MRI views, a re-sliced coronal view along the length of a depth electrode shaft, and the electrode locations in the 3D brain per page of the.pdf. Grids and strips can also be displayed in the 3D brain for surface reconstructions (**Fig 1**). Importantly, the Step 6 output (**Figs 1 and S1**; **Table 1**) can be retrieved without running the 3D brain reconstructions (FreeSurfer, Methods Step 2) if only the MRI views are desired, highlighting the modularity of the pipeline and functionality for different use cases (**S1 Fig**).

To demonstrate the usefulness of the visualization and localization pipeline, we present a case where a patient had undergone pre-surgical evaluation with intracranial leads twice across many years but with slightly different electrode locations (**Fig 2**). In this case, we co-register two different post-operative CTs with different implants to a single T1-weighted MRI scan. We map the electrode locations illustrated relative to the ongoing neural activity recorded by the clinical neurophysiological system (Natus Medical Incorporated) (**Fig 2A**). Often presented in surgical planning conferences, identified seizure activity is compared with the electrode locations in the reconstructions for easy localization (**Fig 2B and 2C**). Mapping the electrode locations from the separate implants reveals if there was regional overlap of epileptiform activity in these two different intracranial investigations in the same participant. Even though the electrodes were different (sEEG vs ECoG strips) and in slightly different locations, the active sites identified as exhibiting epileptiform activity by the clinical team (including board-certified epileptologists) were in a similar area (the brain area near RAH1-2 in the second implant and RSUB5-6 in the first implant; **Fig 2C**). This information could be useful in discussing the epileptiform network as identified in the same participant.

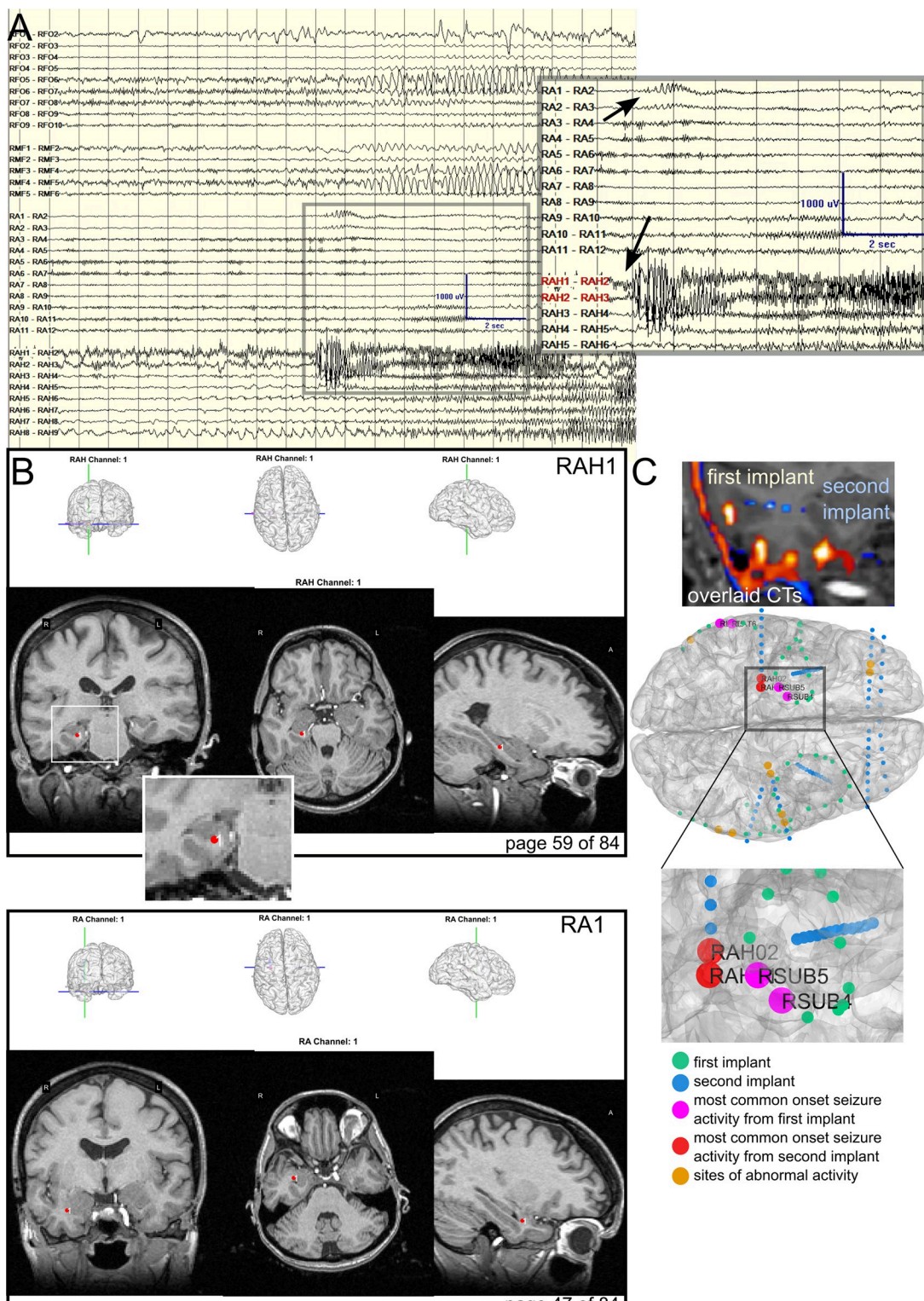

**Fig 2. Electrode localization for mapping seizure onset zones across successive implants. A**. The clinical sEEG recording during seizure onset during the second implant, shown here for comparison's sake. Zoomed image shows the onset of seizure at the channels displayed in panel B (RAH and RA). **B**. Example pipeline output images showing RAH and RA electrode locations relative to three views of the structural pre-op MRI. **C**. How the pipeline can be used clinically to compare electrode placements between old and new implantations on the same patient, including sites which exhibited abnormal activity most frequently per

implant period preceding a seizure. Top: First and second postoperative CTs overlaid on the most recent MRI showing different electrode locations. Blue/cyan is the second implant while red/yellow is the first implant. Bottom: Previously implanted electrodes are green, and the new electrodes are shown in blue. Sites which showed seizures onsets during the first implant period are magenta and during the second implant period are red. Visualizing implanted electrode locations in the structural MRI.

An important step is to validate our co-registrations and visualization output. For this step, we confirm that the locations of the contacts, when plotted on the pre-op MRI, were in concordance with the locations identified using other pipelines. For N>100, clinical teams, led by epileptologists, neurophysiologists, and neurosurgeons, ran separate reconstruction pipelines for making visual representations of the electrodes in the brain, including using Brainstorm, iELVis, or LeadDBS [21,28,29]. Our visuals are always compared to these other reconstructions, and for all cases, we had parity in the location on the MRI scan. Additionally, we have output from the pipeline from 2 naïve users who had no experience with these types of protocols or processes (**S2 Fig**). Not only did their localization match with an experienced user (DJS), but output from the two users matched each other in these steps, with average absolute differences pre contact at 0.97±0.55 mm (across all three RAS axes and contacts) and an average root mean square error relative to the experienced user (DJS) of 0.96 (for naïve user 1) and 0.92 (for naïve user 2; **S2 Fig**). The output from their execution of the protocol demonstrates further the internal consistency and reproducibility of Steps 1–6 of this protocol.

## Electrode localization relative to brain regions

There are a large number of ways to visualize electrode locations in the brain [1,19–30,75,76]. Some pipelines involve transforming the brain regions and electrode locations into a common map (MNI) and visualize activity in a single brain [1,28,29]. Other approaches involve projecting activity onto the surface or onto an inflated (flattened) cortical map [75,76]. Further, when transforming the patient brain into a common atlas, the regions of interest can be represented as voxels using a volumetric approach or as surfaces using the surface-based approach. Typically, surface-based approaches are used for labeling cortical areas because of the inherent variability across cortices, and volume-based approaches are better suited to more homogeneous subcortical labeling [60]. As such, we developed two separate pipelines, one mostly surface based and one purely volume based, for localizing electrodes to brain regions in commonly used atlases per electrode contact or pairs of electrodes (**Figs 3 and 4**). The two methods are 1) the Electrode Labelling Algorithm (ELA) using the Multi-Modal Visualization Tool (MMVT; **Fig 3**) also briefly described elsewhere [6,30,40] and 2) Electrode Volume Labeling (EVL; **Fig 4**), described here for the first time [39].

The ELA is a combined volume and surface approach which involves sampling the volume of space around electrode contacts to identify the nearest surface reconstructed from FreeSurfer, arriving at a probability that the electrode location is near that brain region surface (**Fig 3**). As such, for each contact, there can be multiple probability values above zero of that contact being in white matter or intersecting with the nearest grey matter surface (**Fig 3**). In our current approach, we take the highest probability value in the non-white matter labels to determine the brain region location for each contact (**Fig 3**), though further information is accessible in the ELA output [40]. Finally, after identifying the maximal grey matter probabilities per contact, we can then color code the contacts with the same labels in a common brain with the same atlas (e.g. with the DTK atlas; **Fig 3F**). We provide this code in our pipeline (https://github.com/Center-For-Neurotechnology/Reconstruction-coreg-pipeline) for use. ELA is currently incorporated into the MMVT-lite pipeline for processing the data but can be

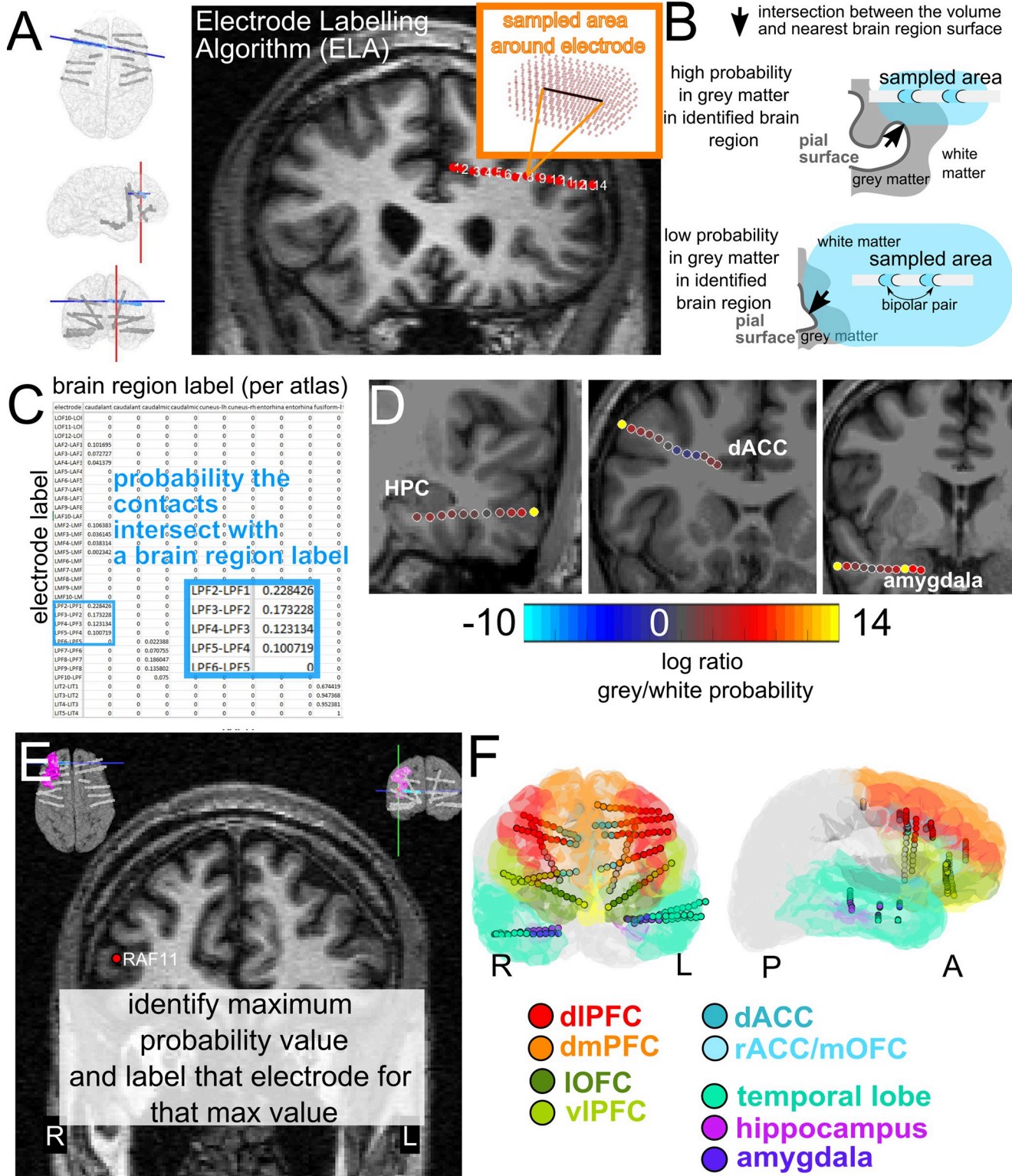

**Fig 3. The electrode labelling algorithm. A**. The electrode labeling algorithm (ELA) involves identifying, in a reconstructed MRI, the location of a depth electrode (red dots overlaid on the MRI, black line in inset) and all the surrounding voxels such that their distance from the electrode is smaller than a given threshold area (red). **B**. Illustration of the intersection of the volume relative to the labelled cortical or subcortical surface created from the FreeSurfer output, indicating contacts with low and high probabilities of being in that brain region. **C**. Example spreadsheet output from ELA of probabilities in contact volumes intersecting with surfaces. **D**. Electrode locations color-coded by log probability of being in nearest grey matter/white matter. **E**. The

electrode brain region labelling approach involves labelling the electrode location relative to the highest grey matter (non-white matter) probability. **F**. Electrodes color coded to brain region locations with the corresponding color-coded surfaces exported from FreeSurfer.

used separately [40]. Not only can we identify the brain region of each contact, but the pipeline also provides localizations as probabilities for better statistical rigor (and supplies the maximum likelihood value). This localization is now applied to almost all research projects performed by our lab as well as other researchers [6,11,13,14,18,31,34–36,38].

The second method developed de novo for this protocol, Electrode Volume Labeling (EVL), uses a purely volumetric approach (**Fig 4**). The steps involve exporting the volumetric aparc segmentations in FreeSurfer [42] using 3DSlicer [77] into.stl files which can be imported into MATLAB or Python for further mapping. Using MATLAB functions, we then classify electrode locations as within different volumes (including white matter volumes). This approach takes into account the thickness of various structures including grey matter. EVL is likely more relevant to depth electrodes than surface arrays (ECoG grids and strips), which is partly why we use both methods for identifying brain region locations per contact.

We have taken measures to validate our ELA and EVL output, though there is much more room for validation in this space. First, for ELA, we provided the 2D electrode localizations (as visualized with the.pdf output) of 20 patients relative to the MRI images to a neurologist and a psychiatrist with training in identifying brain regions with the brain labels. They confirmed the locations through a visual inspection of the localization. For N>200 patients, we have run ELA and found that the locations detected automatically match with the locations found in the MRI upon visual inspection. A limitation here is that these are not trained neuro-radiologists, though we consider these checks be a meaningful indication of accuracy. Further, in comparing EVL versus ELA, we find the methods for identifying brain region per contact are generally consistent between methods, with differences in identification of contacts which are fully in white matter versus grey matter which is may be expected with a volume versus surface based approach (N = 23).

## Electrode characterization and measurements relative to anatomical structures

In parallel to electrode localization relative to brain regions, there are a number of different measures which could be relevant to further study, including localization relative to grey and white matter, contact size, and contact spacing. The organization of the FreeSurfer folders and RAS coordinates allows us to perform automatic calculations of these metrics using basic MATLAB functions without additional toolboxes. An essential step for identifying electrode locations relative to brain regions involves algorithmically and automatically labeling electrode contacts in the patient's native space using purely anatomical landmarks. These measurements include locations relative to anatomical features such as the nearby grey matter, the nearest grey-white boundary, whether contacts are in white or grey matter, or the angles relative to the nearest cortical column. Some of the measures involve volumetric localization (**Fig 4B**). Other measures involve calculating the closest points (vertices) of the nearest imported brain surfaces (**Fig 4C and 4D**). As this approach allows across-patient comparisons automatically, we are able to calculate coverage of electrode contacts in grey matter, white matter, subcortical regions, and outside the brain for different depth trajectories (**Fig 4E**). A depth trajectory is where the distal tip of the sEEG electrode is aimed during implantation. Along this trajectory, we then calculate the number of contacts classified as in grey matter, white matter, subcortical regions, and outside brain using volumetric measures (see **Methods** and **protocol** [39]).

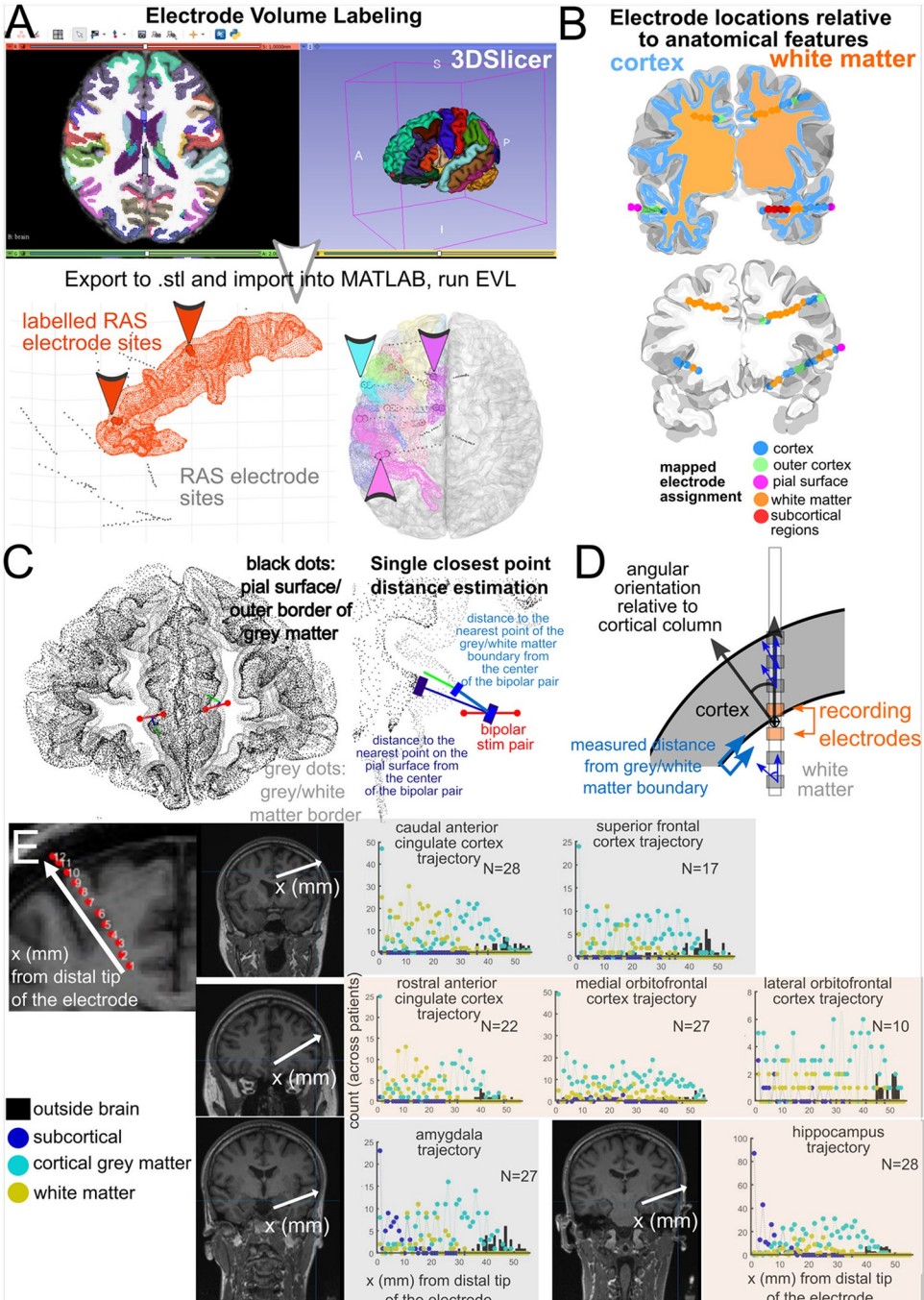

**Fig 4. Electrode Volume Labelling and electrode location measurements. A**. Electrode Volume Labelling (EVL) steps, which involve exporting the aparc+aseg volume brain region labelling from FreeSurfer (such as from the DKT 40 map) to.stl files, which can be imported into MATLAB. Next, electrode contacts are labelled per brain region if they are contained within these volumes, which can contain the thickness of the cortical volume. **B**. Volumetric categorization of electrode contacts as white matter, cortex, subcortical structures, and outside the brain. **C**. Euclidean measurements of distances between the recording contacts and the nearest anatomical features. **D**. Further measurements of electrode contacts relative to different features of the nearest cortical structure. **E**. Categorization of electrode locations relative to neuroanatomical structures (cortical grey matter, subcortical structures, white matter, and outside brain) for depth electrodes with different ending trajectories for different brain regions. Counts of electrode locations relative to different classifications are across individuals (up to 27 patients).

Across one data set (N = 27), the location of electrodes in white matter versus cortical grey matter varies depending on the depth electrode trajectory (**Fig 4E**). This information may be useful for designing novel electrodes for localizing grey matter along common trajectories [78–80]. Further, this localization information and measures of electrode location is instrumental for examining how anatomical features alter neural responses to stimulation or for seizure localization (**Fig 5**) [6,14].

## Use-case: Across-patient mapping, data standardization, and data sharing

Aside from the initial co-registration and electrode localization, the remaining steps in electrode localization, brain region labeling, and mapping are automatic, including measurements of electrode locations relative to neuroanatomical markers (**Figs 4E** and **5A**). Therefore, across patients, we are able to color code contacts paired to brain regions across participants (**Fig 5A**) and summarize counts of contacts per brain region to identify coverage in a data set (**Fig 5B**), allowing for standardization across data sets.

Relevant to this point, as many journals and funding agencies now require the data be shared, we implement de-identification (defacing) and data reformatting into the iEEG BIDS format (see **Methods** and **protocol** [39]) following FAIR practices [72] to ensure we can safely share data following Open Science practices such as on the Data Archive BRAIN Initiative (DABI; https://dabi.loni.usc.edu [81]). At this archive, we share two imaging data sets with their various outputs from the pipeline (https://dabi.loni.usc.edu/dsi/R01NS062092). We have already shared a large intracranial data set using this pipeline (N = 52; [13,14,35]; https://dabi.loni.usc.edu/dsi/W4SNQ7HR49RL). Further electrode localization steps described in the protocol can also be included in the file output in a standardized and well-documented approach (see **protocol** [39]).

## Use-case: Planning for neuromodulation / stimulation

In addition to its clinical uses, this pipeline is also valuable for researchers. For example, this pipeline has been used in understanding how stimulation alters brain activity as well as probe activity in different states such as during sleep and under general anesthesia (**N = 82,** [11,13,14,33–36,38]), where the reconstructed localizations were used to inform selecting stimulation sites. Localizing stimulation sites is important because a key facet of stimulation is understanding the distance between stimulation sites and the brain areas that are targets for stimulation [13,14]. As such, simple calculations of Euclidean distance are crucial for understanding how the brain responds to stimulation (**Fig 5C**). Using the surfaces, localizations, 3D output, and measurements (**Fig 4D**), we can precisely show where and how stimulation will impact a certain area based on the stimulating contact's location and orientation relative to the grey-white matter boundary (**Fig 5C**). We found that stimulation responses vary depending on the stimulation location relative to grey and white matter [14]. We found that stimulation in white matter induced larger distant effects while stimulation closer to the boundary between grey and white matter induced larger local responses (**Fig 5C**) [14]. These metrics and relationships with stimulation may have implications for the use of neuromodulation for therapeutic use and understanding brain connectivity, which is dependent on this pipeline [14].

## Use-case: Localizing and characterizing seizures

Another project set which has relied on this pipeline involves seizure electrophysiology (**Fig 5D**). Understanding the epileptogenic network and the role of different structures in seizure onset, spread, and termination is critical to understanding the underlying mechanisms of epilepsy [6,32]. One way to study seizures is to differentiate them based on their region of

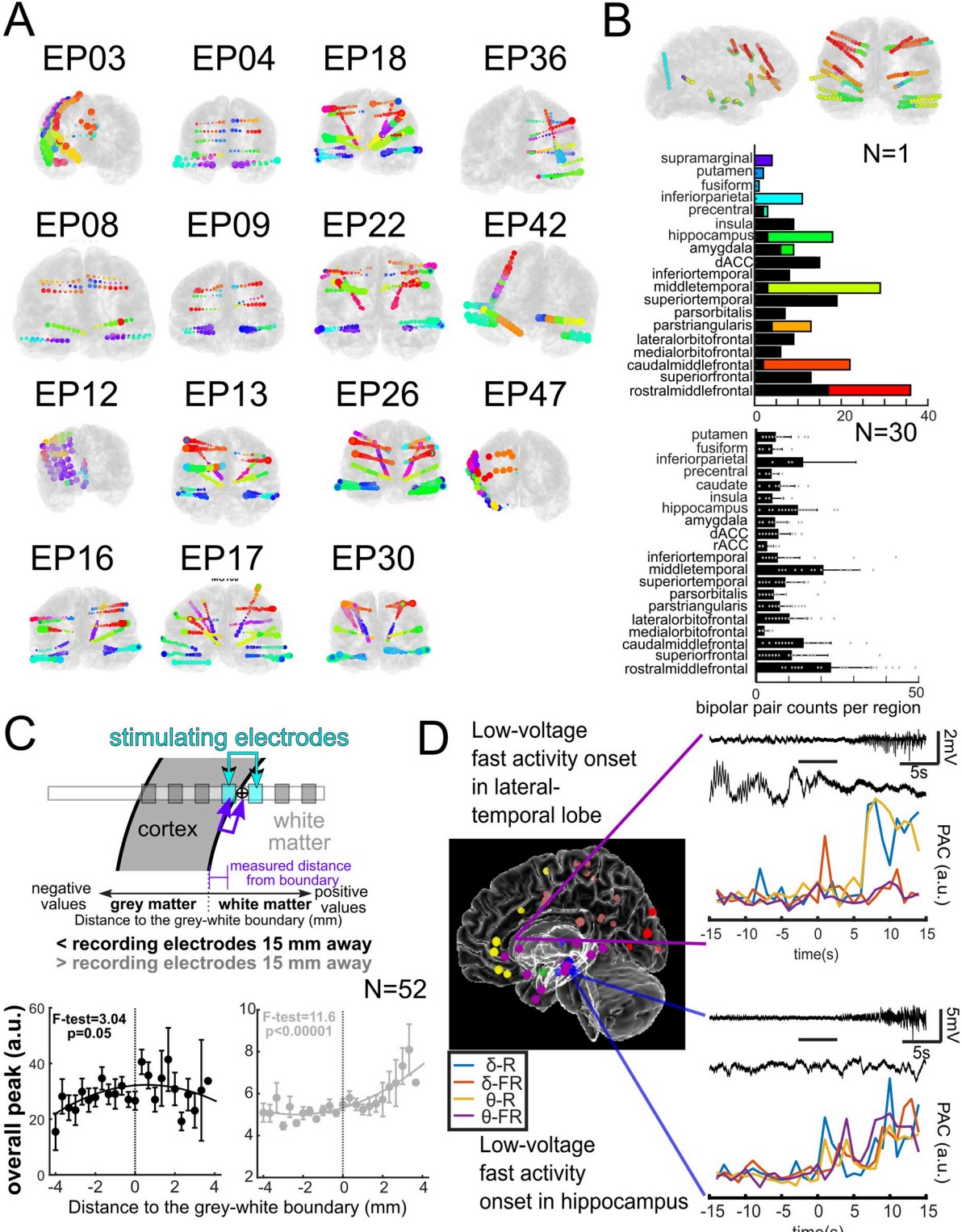

**Fig 5. Pipeline research use cases for stimulation and seizure onset. A.** Electrode localization and labelling across patients. The color coding indicates common labels across participants (designated EPXX) for implanted leads. **B**. The pipeline allows automatic identification of electrode coverage across patients per brain label. Contact counts for different brain regions in individual participant (top) and across multiple participants (bottom, N = 30). **C**. Demonstration of how stimulation electrode location relative to the grey-white boundary can affect responses in local contacts (less than 15 mm away) and in distant contacts (greater than 15 mm; [14]). **D**. Electrode labeling algorithm

(ELA) used to determine the location of electrodes during seizure onset (colored dots represent seizure onsets). Using ELA output, phase-amplitude coupling was analyzed and found to have different dynamics in different brain regions. δ: Delta; θ: Theta; R: Ripple; FR: Fast ripple; PAC: Phase-amplitude coupling; a.u.: Arbitrary unit [6].

onset and the regions to which they spread. Understanding these pathways provides insights into seizure dynamics and how they may eventually be disrupted. For example, we have studied seizures recorded from 43 patients, and we used ELA to determine the brain region in which each electrode registering the seizure onset was most likely to be located in these patients. The seizures were classified based on (a) their electrographic pattern (e.g., low-voltage fast activity) at the onset and (b) the region (e.g., hippocampus, lateral temporal, etc.) from which the seizures originated. The phase-amplitude coupling (PAC) analysis showed that seizures originating from different regions might have more distinct PACs, and therefore the region of onset may distinctively impact the dynamics of seizure initiation. For instance, seizures with low-voltage fast activity at their onset have different PAC dynamics if they originate in the hippocampus versus lateral temporal regions (**Fig 5D**) [6].

## Limitations

This methodology and pipeline has a number of inherent limitations which users should be aware of before using this overall approach. First, the manual step of electrode picking can present challenges such as poor contrast, an overwhelming imaging artifact from the electrodes, and low-resolution imaging. Therefore, it is not entirely surprising that the inconsistencies between electrode contact distances presented here can show up in the localizations. One approach to handle this discrepancy and to validate the localization would be to utilize algorithms to automatically detect the 'bright spots' of a postoperative CT scan and use this information to identify the centers of the electrode contacts, an approach utilized by some software packages [19,20,29]. We do not apply this approach here, but it could be done in future work. A second way to validate the electrode localizations would be to have multiple users localize the contacts, providing cross-user validation (which we have done previously, **S2 Fig**) or compare reconstructions across software packages, though this latter step could be outside the scope of the current study.

Secondly, because this is not a single software package and we make use of multiple software packages (e.g. FreeSurfer, Blender, MATLAB) as well as rely on the command-line for execution of the steps in the pipeline, there is no unifying graphical user interface for controlling all aspects of the pipeline. Therefore, some knowledge of how to navigate between directories through the command-line and how to safely manipulate their contents is required. We reference guides and give tips in our protocol to assuage such fears. Nevertheless, this may be off-putting to potential users who would ideally like to control everything from a single application (such as MATLAB), to potential users with no previous experience controlling an operating system from the command-line, or to potential users who would feel more comfortable controlling everything from a single graphical user interface (such as Lead-DBS or Brainstorm [28,29]).

Thirdly, because the methodology entails the execution of commands in the bash shell—bash being a Unix shell—and preprocessing of the images is handled by FreeSurfer—an application that is only available for Mac OS and Linux OS—not every user or organization will be able to immediately take advantage of this pipeline. While we have suggested running the pipeline through an Ubuntu virtual machine should a Windows user be interested in executing the pipeline, this may be off-putting to people with no prior experience installing or working within virtual machines.

Finally, one major limitation in any electrode localization approach is that the post-operative scans and localizations are only 'snapshots' of the electrode locations. Tissue shifts during and after implant could result in some amount of error. As such, these electrode localizations are best-guess approximations and any analyses and interpretations should keep this issue in consideration. We recommend using the most up-to-date post-operative imaging in order to avoid this problem. The associated brain shifts with ECoG electrodes are a further example of how localization error can propagate as the methodology, as described above, is prone to error or requires assumptions.

This pipeline is an attempt to demonstrate a longstanding collection of methodologies that take advantage of the strengths of different software packages generating all the output described above, with alternatives given along the way (**S1 Fig**). While each software piece may or may not be the perfect tool for each job, we find parts of the pipeline are sufficient for many use cases, and that there are alternatives in part or in whole to our pipeline. The comparison table (**Table 2**) and software pipeline with alternative options (**S1 Fig**) are quick guides on which of the popular software packages may best assist or even substitute for our pipeline methodology for a given feature.

### Future directions

In the future, we anticipate the implementation and possible incorporation of new analysis pipelines into this overall pipeline. This includes white matter tract mapping using such approaches as TRACULA [82,83]. Since seizure disorders like epilepsy are understood to be disorders of the network [84], having more information about how these networks are constructed would be an invaluable tool. This can be used by research teams to answer questions regarding white matter connectivity and its relation to seizure disorders.

Another example of a future tool to add to this pipeline is the inclusion of additional subcortical atlases used in pipelines like LeadDBS [29]. This is a pipeline similar to ours, but where we focus on cortical areas, LeadDBS focuses on subcortical areas with an emphasis on the thalamus. The combination or integration of these two pipelines would also be a major advance for both clinicians and researchers, as both cortical and deep brain structures could be accurately localized and visualized.

While no pipeline like this will be perfect for all use cases, this protocol represents a reproducible pipeline followed by multiple users and applied in hundreds of instances for reconstructing a patient's brain, visualizing implanted electrodes, algorithmically identifying the brain region for each implanted electrode, and organizing all this data into a sharable format. As this type of data becomes more common and widely used, we hope that each of these facets will become more robust, more user-friendly, and more clinically and scientifically impactful.

### Associated content

dx.doi.org/10.17504/protocols.io.5qpvornedv4o/v2.

### Supporting information

**S1 Fig. Flowchart showing the pipeline steps with required software.**
(TIF)

**S2 Fig. Naïve users output for sub-0t3i compared to experienced user (DJS).**
(TIF)

**S1 File. Protocols.io pipeline publication PDF.**
(PDF)

## Acknowledgments

We would like to thank Olivia Gawel, Devang Sehgal, Aniruddha Shekara, and Roshni Khatri for their incredible work testing the feasibility of this pipeline. We would also like to thank Rina Zelmann and Brian Coughlin for their feedback on this manuscript. Finally, we would like to thank most of all the patient-participants with whom we work for volunteering their time and energy to research during their clinical care.

## Author Contributions

**Conceptualization:** Angelique C. Paulk.

**Data curation:** Daniel J. Soper, Dustine Reich, Alex Ross, Noam Peled, Angelique C. Paulk.

**Formal analysis:** Angelique C. Paulk.

**Methodology:** Daniel J. Soper, Noam Peled, Angelique C. Paulk.

**Resources:** Daniel J. Soper, Alex Ross.

**Software:** Daniel J. Soper, Angelique C. Paulk.

**Supervision:** Alex Ross, Pariya Salami, Sydney S. Cash, Ishita Basu, Noam Peled, Angelique C. Paulk.

**Validation:** Daniel J. Soper.

**Visualization:** Daniel J. Soper, Dustine Reich, Pariya Salami, Angelique C. Paulk.

**Writing – original draft:** Daniel J. Soper, Dustine Reich, Angelique C. Paulk.

**Writing – review & editing:** Daniel J. Soper, Alex Ross, Pariya Salami, Sydney S. Cash, Ishita Basu, Noam Peled, Angelique C. Paulk.

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
