## [Decision Letter · Decision Letter 0]

26 Jan 2023

PONE-D-22-33193Modular Pipeline for Reconstruction and Localization of Implanted Intracranial ECoG and sEEG ElectrodesPLOS ONE

Dear Dr. Soper,

Thank you for submitting your manuscript to PLOS ONE. After careful consideration, we feel that it has merit but does not fully meet PLOS ONE’s publication criteria as it currently stands. Therefore, we invite you to submit a revised version of the manuscript that addresses the points raised during the review process.

Your manuscript is of interest and certainly a welcome contribution to efforts at reproducible neuroscience. Reviewers raised a number of concerns/suggestions, that I ask to troughtfully consider and address.

In particular, address the concerns regarding validation, overal accuracy and limitations. Please clarify if/which parts of the protocol are newly introduced. Note that mere standardization of procedures already known is acceptable, but the distinction must be made clear to the reader.

Please diregard any comments regarding novelty and duplicate publication on protocols.io. They don't apply for your contribution.

We look forward to receiving your revised manuscript.

Kind regards,

Federico Giove, PhD

Academic Editor

PLOS ONE

Journal Requirements:

2. Our internal editors have looked over your manuscript and determined that it is within the scope of our Reproducibility and Replicability in Neuroscience and Mental Health Research Call for Papers. The Collection will encompass a diverse and interdisciplinary set of protocols and research articles adhering to transparent and reproducible reporting practices in the areas of clinical psychology, psychiatry, mental health, and neuroscience. Additional information can be found on our announcement page: https://collections.plos.org/call-for-papers/reproducibility-and-replicability-in-neuroscience-and-mental-health-research/. If you would like your manuscript to be considered for this collection, please let us know in your cover letter and we will ensure that your paper is treated as if you were responding to this call. If you would prefer to remove your manuscript from collection consideration, please specify this in the cover letter.

3. Please amend your current ethics statement to address the following concerns:

a) Did participants provide their written or verbal informed consent to participate in this study?

5. Thank you for stating the following in the Funding Section of your manuscript: 

"Support included Fonds de Recherche Santé Québec (FRSQ) postdoctoral fellowship to PS, Tiny Blue Dot foundation to SSC, DJS, DR, and ACP, SSC was funded by NIH grants NINDS R01- NS062092, 1K24NS088568, R01-NS079533, R01-NS072023, and Massachusetts General Hospital Executive Committee on Research (MGH-ECOR). Some of this research was sponsored by the U.S. Army Research Office and Defense Advanced Research Projects Agency (DARPA) under Cooperative Agreement Number W911NF-14-2-0045 issued by ARO contracting office in support of DARPA’s SUBNETS Program. The views and conclusions contained in this document are those of the authors and do not represent the official policies, either expressed or implied, of the funding sources."

"Support included Fonds de Recherche Santé Québec (FRSQ), https://frq.gouv.qc.ca/sante/, postdoctoral fellowship to PS.

Tiny Blue Dot foundation, https://www.tinybluedotfoundation.org/, to SSC, DJS, DR, and ACP. SSC was funded by NIH grants NINDS R01- NS062092, 1K24NS088568, R01-NS079533, R01-NS072023, and Massachusetts General Hospital Executive Committee on Research (MGH-ECOR). Some of this research was sponsored by the U.S. Army Research Office and Defense Advanced Research Projects Agency (DARPA), https://www.darpa.mil/, under Cooperative Agreement Number W911NF-14-2-0045 issued by ARO contracting office in support of DARPA’s SUBNETS Program. The views and conclusions contained in this document are those of the authors and do not represent the official policies, either expressed or implied, of the funding sources. The funders had and will not have a role in study design, data collection and analysis, decision to publish, or preparation of the manuscript."

7. Your ethics statement should only appear in the Methods section of your manuscript. If your ethics statement is written in any section besides the Methods, please move it to the Methods section and delete it from any other section. Please ensure that your ethics statement is included in your manuscript, as the ethics statement entered into the online submission form will not be published alongside your manuscript. 

Reviewers' comments:

Reviewer's Responses to Questions

**Comments to the Author**

1. Does the manuscript report a protocol which is of utility to the research community and adds value to the published literature?

Reviewer #1: Yes

Reviewer #2: Yes

Reviewer #3: Yes

Reviewer #4: No

2. Has the protocol been described in sufficient detail?

To answer this question, please click the link to protocols.io in the Materials and Methods section of the manuscript (if a link has been provided) or consult the step-by-step protocol in the Supporting Information files.

The step-by-step protocol should contain sufficient detail for another researcher to be able to reproduce all experiments and analyses.

Reviewer #1: Partly

Reviewer #2: Partly

Reviewer #3: Yes

Reviewer #4: Partly

3. Does the protocol describe a validated method?

Reviewer #1: No

Reviewer #2: No

Reviewer #3: Yes

Reviewer #4: Yes

4. If the manuscript contains new data, have the authors made this data fully available?

Reviewer #1: Yes

Reviewer #2: No

Reviewer #3: Yes

Reviewer #4: N/A

**5. Is the article presented in an intelligible fashion and written in standard English?**

Reviewer #1: Yes

Reviewer #2: Yes

Reviewer #3: Yes

Reviewer #4: Yes

6. Review Comments to the Author

Reviewer #1: The authors provide an overview of their software suite to enable intracranial electrode localization, visualization, and anatomical labeling. There are a lot of packages that do similar things, but theirs is particularly notable for the large number of patients it has been used on, providing output in BIDS format, and the large number of optional outputs (like each electrode's distance to the nearest gray/white junction). This will be a useful piece of software. But a few more things should be done with the paper first.

Nothing is currently done to validate any of the localizations. What is the gold standard here, and how do we know the software is performing correctly? Other papers have done things like compare expert anatomical labeling to the software (like a neuroradiologist) for a subset of patients.

A lot of the derived measurements, like Euclidean distances between electrodes, depend on accurate localization of the electrodes. But you can tell from the images that the SEEG contacts are probably not well localized. These linear devices have fixed inter-contact distances that can't be altered--the electrodes can bend but not compress or stretch. Fig. 4, for instance, shows several electrodes where the contacts are not colinear or on the same curved trajectory, but look "jaggedly" arranged. What do the authors make of this apparent error in the mapping?

One problem with all these software packages is that they lend a sense of exactitude to something that is not exact. This paper needs to spend some time talking about its limitations. Some examples are registration/fusion error, "snapping" grid electrodes to the brain surface (lots of error there), manually localization of SEEG contacts, and so on. Discussing these limitations is mandatory. Optional: To make a really great paper, the authors could assess the magnitude of these potential errors and quantify them. That way, when the software spits out a list of euclidean distances, it could also spit out a confidence interval, for example. This is similar to how they are ascribing the probability of an anatomical label. Steps like this would be amazing and really help the field.

The authors should do a more comprehensive job of listing the alternative software packages, their pros/cons, and how the current software package is different. Perhaps a table.

Some estimate of the time required to perform these steps is needed.

Some estimate of what kind of computer is required to perform these steps is needed. What are the minimum requirements?

More minor comments follow:

In lines 105-106, the authors equate grids and strips to ECoG and as something separate from SEEG. Electrocorticography means an electrical picture/drawing ('-graphy') from the cortex, which would describe most of SEEG too.

Line 175: Ad-Tech depth electrodes are 0.86 mm at the smallest, not 0.8 mm.

I did not review the Protocols.io document in detail, but it's wonderful that the authors included this supplement!

Step 1 needs more details on the expected imaging formats (NIFTI, DICOM, etc) and if there are any restrictions on DICOM headers (can they be anonymized?). Are there expected file names to identify each sequence? What are the minimum imaging resolutions required? What if there are multiple instances of the same sequence, like 2 versions of a post-op CT, one on post-op day 0 and one a few days later? Can both be used? Just in general more information is needed in this section, more than the single sentence there now

Lines 209-210 discuss needing an algorithm to map surface electrodes to the brain surface. More detail is needed here. Which algorithm is used and why that particular algorithm and not alternatives?

Reviewer #2: Summary: The authors describe a pipeline for localization of intracranial electrodes, implanted for diagnosing medically refractory epilepsy. The pipeline relies on basic functions from numerous other software packages. While it is potentially useful to have a pipeline such as this down on paper, the advances beyond the functions borrowed from other packages do not warrant the promotional style of writing. This reviewer is also unsure why this is being considered for publication at a journal, given that the protocol is already published with a DOI on protocols.io. The work doesn’t meet several of the criteria for publication as an article in PLoS ONE (original research not published elsewhere), but apparently these criteria do not apply for protocols? The major novel contribution of this work is the code that de-identifies images and formats them for BIDS.

Minor critiques:

- Line 116: It may be clearer to replace ‘implantation’ with ‘implanted electrodes’

- Line 152: ‘we remain in patient space’ should be reworded.

- Line 154: ‘alternations’ should be ‘alterations,’ I believe.

- The protocol relies on bash commands in linux, which limits its widespread utility.

- The protocol states the importance of the need to quickly generate co-registrations, but then uses recon-all in freesurfer, which takes tens of hours to run.

Major critiques:

- My major issue with this manuscript is stylistic. The manuscript too promotional in style. This is especially relevant given that the manuscript describes a protocol that is heavily based on functions from other widely used software packages.

- There are no example data provided to test the protocol. Given how heavily this work is based on other folks research, the least the authors can do is make it easy to test the protocol.

Reviewer #3: In this paper, a protocol proposed for localizing electrodes implanted in the brain which is accessible to multiple skill levels and modular in execution.

1) Step 5: Snapping the Electrode Grid

Please explain briefly in the text how the shift and compression in the brain as a result of the surgery will be compensated. It is not clear.

2) In Fig. 2, the electrode locations from the separate implants were mapped. Are you going to say that the method could identify electrodes sites which show epileptiform activity and consistent with what identified by board-certified epileptologists. From the signal recorded from each electrode, we could identify the seizure onset. Using the proposed algorithm, we could identify the electrode locations. What is consistency here? Different onset seizure locations were identified during different implants. How can we say consistent? Please clarify.

3) Fig. 2A correspond to what implant; first or second. Please clarify. Clarify what is RAH1 and RA1 in Fig. 2.

4) It was suggested that the region of seizure initiation may be more informative of different dynamics than the electrographic pattern at seizure onset (line 478). Please clarify how? How did you find it is more informative? What is PAC in Fig. 5? It seems to be phase amplitude coupling. It should be clarify in the Figure caption.

5) Fig. 4: Please clarity what is the right MRI plot in Fig 4E.

6) Define the abbreviations used in the text and figures.

7) The steps to identify the electrode position were presented in the paper. It would be very useful to provide the steps for installing the pipeline code and the required package and software. Does the pipeline code collect all the necessary codes in a single code?

Reviewer #4: Authors of “Modular pipeline for reconstruction and localization of implanted intracranial ECOG and sEEG electrodes”, describe a modular and extendable pipeline for processing, visualizing and analyzing intracranial data in human subjects – both electrocorticography strips/grids and stereoencephalography. Their manuscript proposes the combination of several long-established and some new programmatic routines to accomplish the processing/visualization steps. Importantly the proposed approach includes the ability to generate sharable data in NIH approved formats. This is a truly admirable effort to address many challenges for these types of data. My 3 primary concerns with the manuscript are 1) it was difficult to ascertain which aspects of the pipeline were derived from established software (seemed like most of the pipeline) versus the novel algorithms/programs generated by the authors, 2) there are many efforts that have attempted to address this challenge – maybe this would be more relevant for software only journal or as additional methods for a scientific manuscript that relies on this method 3), the method will be hard to follow; authors make an allusion to Lead-DBS as a comparison – Lead-DBS is a stand-alone Matlab program that does not require any additional downloads or programs and still has a highly active Slack channel for issues; secondly as a guide to the extent that this kind of work should be documented see Stolk et al., 2018 Nature Protocols (which is still not trivial to follow/replicate).

7. PLOS authors have the option to publish the peer review history of their article (what does this mean?). If published, this will include your full peer review and any attached files.

Reviewer #1: No

Reviewer #2: No

Reviewer #3: **Yes: **Abbas Erfanian

Reviewer #4: No

---

## [Author Response · Author response to Decision Letter 0]

2 May 2023

Journal Requirements:

1. Please ensure that your manuscript meets PLOS ONE's style requirements, including those for file naming. The PLOS ONE style templates can be found at: 

 We have gone through and corrected the manuscript so that it meets the standards set by the PLOS ONE style guides.

2. Our internal editors have looked over your manuscript and determined that it is within the scope of our Reproducibility and Replicability in Neuroscience and Mental Health Research Call for Papers. The Collection will encompass a diverse and interdisciplinary set of protocols and research articles adhering to transparent and reproducible reporting practices in the areas of clinical psychology, psychiatry, mental health, and neuroscience. Additional information can be found on our announcement page: https://collections.plos.org/call-for-papers/reproducibility-and-replicability-in-neuroscience-and-mental-health-research/. If you would like your manuscript to be considered for this collection, please let us know in your cover letter and we will ensure that your paper is treated as if you were responding to this call. If you would prefer to remove your manuscript from collection consideration, please specify this in the cover letter.

 We thank the editors very much for pointing out this call for papers which fits our protocol so well. We have indicated our desire to be considered for this “Reproducibility and Replicability in Neuroscience and Mental Health Research” collection in our cover letter.

3. Did participants provide their written or verbal informed consent to participate in this study?

The Ethics Statement was amended to clarify that the full consent process is both verbal and written in that the initiation of the consent process starts with a conversation with final consent given through a written form.

4. We note that the grant information you provided in the ‘Funding Information’ and ‘Financial Disclosure’ sections do not match. When you resubmit, please ensure that you provide the correct grant numbers for the awards you received for your study in the ‘Funding Information’ section.

 We have checked the grant numbers and ensured that they are correct on re-submission.

5. We note that you have provided funding information that is not currently declared in your Funding Statement. However, funding information should not appear in the Acknowledgments section or other areas of your manuscript. We will only publish funding information present in the Funding Statement section of the online submission form. Please remove any funding-related text from the manuscript and let us know how you would like to update your Funding Statement. Please include your amended statements within your cover letter; we will change the online submission form on your behalf.

 We have removed the Funding section in the manuscript and will include the amended statement in the Cover Letter.

The data is available for a large subset of 52 patients (https://dabi.loni.usc.edu/dsi/W4SNQ7HR49RL), but we do not include all the outputs from the pipeline. We have therefore added data for two participants to a single repository (https://dabi.loni.usc.edu/dsi/R01NS062092) and have indicated as such in the manuscript.

7. Your ethics statement should only appear in the Methods section of your manuscript.

The ethics statement has been moved into the Methods section and deleted from any other section.

Reviewer 1

The authors provide an overview of their software suite to enable intracranial electrode localization, visualization, and anatomical labeling. There are a lot of packages that do similar things, but theirs is particularly notable for the large number of patients it has been used on, providing output in BIDS format, and the large number of optional outputs (like each electrode's distance to the nearest gray/white junction). This will be a useful piece of software. But a few more things should be done with the paper first.

We thank the reviewer for their positive comments and valuable input below; however, there is one point which we want to make clear. Specifically, we have attempted to clarify in the manuscript that this is a methodology and pipeline incorporating multiple software packages and is not a single software package. We realize this may be a limitation for some users and have highlighted this point in the updated “Limitations” section in the manuscript. We have found it beneficial to use multiple packages for different purposes to enable flexibility, modularity, and optional outputs which is why this protocol is not an all-in-one software package solution and has been useful for a number of different functions and a large data set over the years (N>260).

1. Nothing is currently done to validate any of the localizations. What is the gold standard here, and how do we know the software is performing correctly? (Other papers have done things like compare expert anatomical labeling to the software (like a neuroradiologist) for a subset of patients.

We thank the reviewer for their excellent point regarding the electrode localizations and labelling, particularly with regard to a gold standard for labeling. At one point, we did ask two experts (a neurologist and a psychiatrist) to examine the output which identified contact locations relative to the electrode localizations overlaid on the MRI brain. They both agreed that the labeling appeared accurate for 20 participants. Further, we developed two different approaches for identifying electrode locations (ELA and EVL) which, by and large, agree in electrode locations in grey matter. Some software packages perform validations by automatically detecting ‘bright’ spots from a postoperative CT and compare these relative to the RAS coordinates. We point to this option as a validation approach but, as we are not, for the most part, developing software in this manuscript, we did not implement this validation approach here. We refer to these approaches in the limitations sections as well as in the results.

We further asked two naïve users to perform electrode localization with the same data set to determine if there is user bias which could alter the locations (S3 Fig) and found across the board agreement between the new users and an experienced user (with average absolute differences pre contact at 0.97±0.55 mm). A final approach for validation would be to compare the electrode locations from post-operative MRIs where possible (which has been done in a subset of participants). This step, though, is the reason for the .pdf document we produce. In the .pdf output document, we are not making any assumptions regarding electrode location labeling per brain region but making it available for clinicians and experts to visually confirm electrode location. 

2. A lot of the derived measurements, like Euclidean distances between electrodes, depend on accurate localization of the electrodes. But you can tell from the images that the SEEG contacts are probably not well localized. These linear devices have fixed inter-contact distances that can't be altered--the electrodes can bend but not compress or stretch.) Fig. 4, for instance, shows several electrodes where the contacts are not colinear or on the same curved trajectory, but look "jaggedly" arranged. What do the authors make of this apparent error in the mapping?

We thank the reviewer for their insightful comment. We agree that the contacts should have some constraints in that the inter-contact distances should not compress or stretch. However, as the manual step of electrode picking can be presented with challenges in that the contrast can be poor, the imaging artifact from the electrodes can be overwhelming, and the imaging quality can be low resolution, it is not entirely surprising that the jitter presented here can show up in the localizations. We have included more of these limitations in the manuscript itself. To get around this, several software packages attempt to include a best fit line (or a single vector as in Brainstorm) or use a 3D model of an electrode (such as in DBS within LeadDBS). In the past, we have also created 3D models of the cylinders of the electrode contacts which then can be modelled. We did not include these steps in this protocol but can if it is of use.

However, we were not as concerned about the jagged appearance as we also recognize there is some amount of error. The hope, then, was that the population-scale measures could reveal relationships between activity and electrode locations considering there will always be some error with these measures (in that the electrode could be in a slightly different location as time progresses). We have included this limitation in the manuscript as well. 

3. One problem with all these software packages is that they lend a sense of exactitude to something that is not exact. This paper needs to spend some time talking about its limitations. Some examples are registration/fusion error, "snapping" grid electrodes to the brain surface (lots of error there), manually localization of SEEG contacts, and so on. Discussing these limitations is mandatory. Optional: To make a really great paper, the authors could assess the magnitude of these potential errors and quantify them. That way, when the software spits out a list of euclidean distances, it could also spit out a confidence interval, for example. This is similar to how they are ascribing the probability of an anatomical label. Steps like this would be amazing and really help the field.

The reviewer brings up an excellent point regarding confidence intervals and the usefulness of communicating the accuracy and precision or each measurement. We have expanded the limitations section of our manuscript to discuss these issues. We also tried to clarify that this manuscript is about sharing a methodology used in relatively large number of patients (N>260) to localize electrodes and less about the software development, particularly the use of the “snapping” grid approach. On this point in particular, there are other papers and methods which better detail this step which we now refer to in the manuscript, included in Table 2 and a detailed software flowchart in S1 Fig. 

 With regard to the idea of identifying the magnitude of the potential errors, we completely agree with the reviewer that producing confidence intervals which identify errors and probabilities would be an excellent study. However, we believe this approach is outside the scope of this manuscript currently as this is a protocol with the purpose of addressing “Reproducibility and Replicability in Neuroscience and Mental Health Research”, not necessarily a software development manuscript with validation. We have tried to further clarify this point throughout the manuscript. 

4. The authors should do a more comprehensive job of listing the alternative software packages, their pros/cons, and how the current software package is different. Perhaps a table.

We thank the reviewer for their excellent suggestion on adding a table. We have included a table in the manuscript (Table 2) as well as flow chart of packages, alternatives, and outputs (S1 Fig). We also hope this table and flow chart figure can be used as further evidence to clarify that this is not a manuscript for detailing a single software package so much as an expandable methodology for addressing our various needs. 

5. Some estimate of the time required to perform these steps is needed. 

For each step in the manuscript, we added an estimated amount of time for each one, and for some we gave clarifying information about level of involvement.

6. Some estimate of what kind of computer is required to perform these steps is needed. What are the minimum requirements?

Thank you for pointing this issue out. We have included a list of requirements for a computer at the beginning of our Materials and methods section as listed below:

Line 210: “In terms of the necessary computational hardware to support this pipeline, the reconstruction steps using FreeSurfer are the most demanding. Therefore, we recommend that the system meet their requirements (https://surfer.nmr.mgh.harvard.edu/fswiki/rel7downloads) as the rest of the pipeline is less computationally intense. There are four main hardware requirements: (1) Intel processor supporting AVX instructions, (2) 8GB of RAM for the reconstruction, or 16GB of RAM for better graphical viewing, (3) a 3D graphics card with its own memory and accelerated OpenGL drivers, and finally (4) around 300MB of free hard drive space per processed subject.”

More minor comments follow:

1. In lines 105-106, the authors equate grids and strips to ECoG and as something separate from SEEG. Electrocorticography means an electrical picture/drawing ('-graphy') from the cortex, which would describe most of SEEG too.

We thank the reviewer for their input. We clarified in the manuscript (seen below) that, for this paper, we use ‘ECoG’ to refer to ECoG grids and strips which lay on the cortical surface, and ‘sEEG’ to refer to ECoG depths which aim for medial cortical/subcortical areas. We chose that separation as sEEG can be recording from subcortical structures as well which is why we separated the two designations.

Line 110: “These surgically implanted electrodes consist of two types: 1) contacts along thin tubes that extend into the brain (also intraparenchymal depths) targeting subcortical brain regions or deep cortical structures, which we call stereotactic electroencephalography (sEEG) electrodes, or 2) grids and strips on silastic sheets that lay on the cortical surface, which we call electrocorticography (ECoG) electrodes.”

2. Line 175: Ad-Tech depth electrodes are 0.86 mm at the smallest, not 0.8 mm.

We have changed the manuscript so that it states the more precise 0.86mm diameter. 

3. Step 1 needs more details on the expected imaging formats (NIFTI, DICOM, etc) and if there are any restrictions on DICOM headers (can they be anonymized?). Are there expected file names to identify each sequence? What are the minimum imaging resolutions required? What if there are multiple instances of the same sequence, like 2 versions of a post-op CT, one on post-op day 0 and one a few days later? Can both be used? Just in general more information is needed in this section, more than the single sentence there now

Thank you for the input. The FreeSurfer software for running recon-all requires an MRI in the form of a NIFTI, DICOMS, or .MGZ images. The header information can be de-identified (and even de-faced), and we do this in our pipeline. For our pipeline, we expect that the MRI NIFTI be called ‘mri.nii’, though this is only a requirement in our MATLAB code for creating the .pdf visuals (step 6). The specifics of this are explained more clearly in our protocol.io submission. The minimum imaging resolution is set by FreeSurfer, so we can find and defer to the requirements by FreeSurfer (as listed in their website). 

The questions concerning post-op CTs are unique to the needs of the user. Using the most recent Post-Operative CT imaging will arguably be more accurate considering brain shift should diminish over time. Our group’s standard operating procedure is to use the most recent CT. To use multiple CTs, you would need to run through Steps 3-7 with each CT and ensure that any subsequent CTs are co-registered accurately with the original structural T1 used with FreeSurfer. This process is in fact what we did for one of our Use Cases which involve mapping implants of different electrodes (separated by years) to the same preoperative MRI to see if there is colocalization of epileptiform activity (Fig 2). There is generally flexibility in this paradigm as long as the images are all registered to a single identified T1-weighted MRI. We included this information below:

Line 249: “A key point is that all the imaging is registered to a single identified pre-operative T1-weighted MRI scan. As such, if multiple CT scans are performed, each can be registered to the single pre-operative T1-weighted MRI scan sequentially such that the coordinates can be mapped to the same space.”

4. Lines 209-210 discuss needing an algorithm to map surface electrodes to the brain surface. More detail is needed here. Which algorithm is used and why that particular algorithm and not alternatives?

Thank you for this input. We would like to point out that there are are many methodologies for this and a number of publications and software packages to do this step. For our pipeline, we happen to use the methodology described by Dykstra et al. (2011) and incorporated into iELVIS out of convenience, as the basic pipeline developed in this lab was used to develop iELVis. However, many years ago, we tested multiple approaches in several patients using both the Yang et al. (2012) and Dykstra et al. (2011) projection methods and found that the Dykstra et al. approach was slightly easier to implement with better results (i.e., less likely to fail, better inter-contact spacing maintenance). Most of the alternative methods we discuss in this manuscript also use the Dykstra et al. projection method. Despite that, we think the reviewer’s suggestion is extremely important, so we have incorporated several references to other methods, if the reader is inclined.

Line 266: “Various researchers have identified algorithms to shift the electrodes to the surface of the brain while constraining the relationships between contacts, though these details are reported and validated elsewhere [21,65,66]. We use the extracted and smoothed surface of the pre-operative MRI as a target for snapping the grids and strips, specifically using the method detailed in Dykstra et al. [5]. This method was selected for its ease of use with the pipeline, but many other pipelines can handle this step as well if desired (Table 2, S1 Fig) [5,21,65,66].”

Reviewer 2

Summary: The authors describe a pipeline for localization of intracranial electrodes, implanted for diagnosing medically refractory epilepsy. The pipeline relies on basic functions from numerous other software packages. While it is potentially useful to have a pipeline such as this down on paper, the advances beyond the functions borrowed from other packages do not warrant the promotional style of writing. This reviewer is also unsure why this is being considered for publication at a journal, given that the protocol is already published with a DOI on protocols.io. The work doesn’t meet several of the criteria for publication as an article in PLoS ONE (original research not published elsewhere), but apparently these criteria do not apply for protocols? The major novel contribution of this work is the code that de-identifies images and formats them for BIDS.

We thank the reviewer for their input. We had hoped that, instead of being under the purview of original research, that this particular manuscript would fall under the category of “Reproducibility and Replicability in Neuroscience and Mental Health Research”. We therefore agree this manuscript is not original research so much as providing a methodology that can have multiple outputs depending on the user need. We attempt to clarify this point at multiple points in this revised manuscript. 

Further, we have attempted, as best we could, to remove any language suggesting a ‘promotional’ style. This protocol and associated manuscript are simply a way to communicate a methodology and pipeline incorporating multiple software packages that has worked for a large data set (N>260) for several years with multiple users. As such, we agree it is not necessarily a novel contribution though, as per the purview of reproducibility, we hope the pipeline or parts of it can be of help to the community, particularly the flexibility and modularity of the protocol, video instructions of the steps involved (now incorporated into the updated protocos.io site), and shared code. 

Minor critiques:

1. Line 116: It may be clearer to replace ‘implantation’ with ‘implanted electrodes’

 Thank you for helping us to clarify this point. We have replaced ‘implantation’ with ‘implanted electrodes’.

2. Line 152: ‘we remain in patient space’ should be reworded.

 We have specified that the images and coordinates are the things which remain in native space. We hope this clarifies our original intention.

3. Line 154: ‘alternations’ should be ‘alterations,’ I believe.

 This has been corrected to ‘alterations’.

4. The protocol relies on bash commands in linux, which limits its widespread utility.

 Though this is mainly a limitation of FreeSurfer, it certainly does affect the accessibility of this pipeline. That being said, it does still work with MacOS, and Windows has integrated Ubuntu terminals which can run FreeSurfer in a virtual environment (Install WSL | Microsoft Learn). Though it will be more difficult for users with only a dedicated Windows machine, there is still a documented path for FreeSurfer and this pipeline’s usage for them.

5. The protocol states the importance of the need to quickly generate co-registrations, but then uses recon-all in freesurfer, which takes tens of hours to run.

 The reviewer makes an excellent point. The pipeline can quickly generate co-registrations and carry out the creation of visualizations without FreeSurfer. FreeSurfer simply allows for the 3D reconstruction and labelling of brain regions. If the user only requires the overlay of the electrode locations onto 2D representations of the MRI, the FreeSurfer step is not needed. To clarify this point, we added language and estimated times-to-complete for the different aspects of the pipeline so that it is clear that using the pipeline with FreeSurfer will take much longer. However, an important point is that the FreeSurfer step could also be performed before the implant surgery using the preoperative MRI scan. Therefore, the 3D aspects of the pipeline can still be run relatively quickly if the FreeSurfer is run before implant. We have added language to clarify that this is our use-case, but that users running through the pipeline after implant will need to budget extra time for 3D aspects. 

 Line 281: “This step is time-consuming and is not necessary if only the 2D visualizations from Step 6 are desired. However, there are different paths for different outputs associated with this pipeline which are detailed both in Table 1 and in two flow charts (Fig 1; S1 Fig). For the output from the “Basic” or “Advanced” pipelines, we find starting Step 2 as soon as possible, even before the initial electrode implant, can offset the amount of time the FreeSurfer pipeline takes.”

Major critiques:

1. My major issue with this manuscript is stylistic. The manuscript too promotional in style. This is especially relevant given that the manuscript describes a protocol that is heavily based on functions from other widely used software packages.

We thank the reviewer for their valuable feedback. We have gone through the manuscript in detail to decrease the amount of promotional statements and, in particular, made it more clear with regard to the limitations of these approaches. We also expanded our Limitations section and included Table 2 and S1 Fig which lists the software packages and useful functions each provide currently including alternatives and relevant pipeline outputs. 

2. There are no example data provided to test the protocol. Given how heavily this work is based on other folks research, the least the authors can do is make it easy to test the protocol.

Thank you for this input. We have included example data to test the protocol which includes pre-operative and post-operative scans for two participants as well as the outputs from the pipeline as useful examples. This data is currently shared on the DABI website (https://dabi.loni.usc.edu/dsi/R01NS062092). Further, pre and postoperative scans as well as example FreeSurfer output along with the RAS coordinates relative to the T1 per participant are already available for a, large, deidentified data set for further tests (N=52; https://dabi.loni.usc.edu/dsi/W4SNQ7HR49RL). 

Reviewer 3

In this paper, a protocol proposed for localizing electrodes implanted in the brain which is accessible to multiple skill levels and modular in execution.

We thank the reviewer for their detailed input and suggestions which we feel improve the manuscript. 

1. Step 5: Snapping the Electrode Grid -- Please explain briefly in the text how the shift and compression in the brain as a result of the surgery will be compensated. It is not clear.

We thank the reviewer for their request. We have added more information in Step 5 regarding how compression in the brain is compensated with this current approach. We also detail some of the limitations of this approach in the manuscript.

2. In Fig. 2, the electrode locations from the separate implants were mapped. Are you going to say that the method could identify electrodes sites which show epileptiform activity and consistent with what identified by board-certified epileptologists. From the signal recorded from each electrode, we could identify the seizure onset. Using the proposed algorithm, we could identify the electrode locations. What is consistency here? Different onset seizure locations were identified during different implants. How can we say consistent? Please clarify.

We clarified our point and removed the term ‘consistent’. We were originally referring to the fact that, even though the recordings were from subdural strips and sEEG depth electrodes in implants separated by years, the same general region (the brain area near RAH1-2 and RSUB5-6) was active and produced epileptiform activity in the same area as identified by the clinical teams. As the electrodes were not placed in the exact same locations, we cannot definitively state that they are in the same spots, but we were highlighting how we could observe similar activity in the same general region. This information can be useful when discussing where the seizure onset zone could be or what areas of the brain are involved in the epileptiform network.

Line 418: “Mapping the electrode locations from the separate implants reveals if there was regional overlap of epileptiform activity in these two different intracranial investigations in the same participant. Even though the electrodes were different (sEEG vs ECoG strips) and in slightly different locations, the active sites identified as exhibiting epileptiform activity by the clinical team (including board-certified epileptologists) were in a similar area (the brain area near RAH1-2 in the second implant and RSUB5-6 in the first implant; Fig 2C). This information could be useful in discussing the epileptiform network as identified in the same participant.”

3. Fig. 2A correspond to what implant; first or second. Please clarify. Clarify what is RAH1 and RA1 in Fig. 2.

Thank you for the points. Fig. 2A refers to the second implant. RAH1 and RA1 are different depth electrodes which are then pictured in B. We have clarified these points in the manuscript and figure legend.

4. It was suggested that the region of seizure initiation may be more informative of different dynamics than the electrographic pattern at seizure onset (line 478). Please clarify how? How did you find it is more informative? What is PAC in Fig. 5? It seems to be phase amplitude coupling. It should be clarify in the Figure caption.

Thank you for this clarifying question. As the reviewer has suggested, the PAC stands for "phase-amplitude coupling". We have now added the abbreviations to the figure legend. Additionally, we have modified the main text to include a summary of the study as in the following:

Line 592: “For example, we have studied seizures recorded from 43 patients, and we used ELA to determine the brain region in which each electrode registering the seizure onset was most likely to be located in these patients. The seizures were classified based on (a) their electrographic pattern (e.g., low-voltage fast activity) at the onset and (b) the region (e.g., hippocampus, lateral temporal, etc.) from which the seizures originated. The phase-amplitude coupling (PAC) analysis showed that seizures originating from different regions might have more distinct PACs, and therefore the region of onset may distinctively impact the dynamics of seizure initiation. For instance, seizures with low-voltage fast activity at their onset have different PAC dynamics if they originate in the hippocampus versus lateral temporal regions (Fig. 5D)[6].”

5. Fig. 4: Please clarity what is the right MRI plot in Fig 4E.

Thank you for pointing this out to us. The right MRI plot in figure 4E was to demonstrate the extent of differences in the trajectory of depth electrodes for similar targets. We attempt to show that different trajectories result in different spreads of electrode localizations in areas such as grey matter, white matter, or outside the brain. We have clarified this in the text and figure. 

6. Define the abbreviations used in the text and figures.

We have added more information and defined abbreviations in the manuscript.

7. The steps to identify the electrode position were presented in the paper. It would be very useful to provide the steps for installing the pipeline code and the required package and software. Does the pipeline code collect all the necessary codes in a single code?

Thank you for the inquiry. We have a detailed explanation of how to download the necessary code in the protocol.io submission made in tandem with this manuscript. We tried to keep all the processing code in the GitHub page made for this protocol pipeline. We also attempted to clarify that this pipeline involves multiple software packages which are dependent on user need and preferences, particularly dependent on the type of output the user requires. We have added a flow chart to better describe these steps in the manuscript and clarify what parts are needed for each step, including dependencies (S1 Fig).

For instance, there are two separate pipelines we are using for identifying electrode location relative to different brain regions and electrode labelling per brain region. In one pipeline, the code uses MMVT for the ELA mapping. In the other pipeline (EVL), the code uses various metrics within MATLAB along with Freesurfer and 3DSlicer for localizing the electrode location. We present code which then brings together these pipelines into a single script which we have run on multiple patients so far (N=23). We have attempted to clarify this point further in the manuscript at multiple points.

Reviewer 4

Authors of “Modular pipeline for reconstruction and localization of implanted intracranial ECOG and sEEG electrodes”, describe a modular and extendable pipeline for processing, visualizing and analyzing intracranial data in human subjects – both electrocorticography strips/grids and stereoencephalography. Their manuscript proposes the combination of several long-established and some new programmatic routines to accomplish the processing/visualization steps. Importantly the proposed approach includes the ability to generate sharable data in NIH approved formats. This is a truly admirable effort to address many challenges for these types of data. 

We thank the reviewer for their positive comments and note of support. 

My 3 primary concerns with the manuscript are:

1) It was difficult to ascertain which aspects of the pipeline were derived from established software (seemed like most of the pipeline) versus the novel algorithms/programs generated by the authors.

We thank the reviewer for making this point. As such, we created a flow chart in the manuscript (S1 Fig) and a table (Table 2) to better outline the parts which were established software versus code generated by ourselves. As this manuscript was meant to lay out a methodology incorporating multiple software packages, we attempt to make this goal clearer in the current manuscript at multiple points. 

2) There are many efforts that have attempted to address this challenge – maybe this would be more relevant for software only journal or as additional methods for a scientific manuscript that relies on this method 

We completely agree there are many attempts to address this challenge, particularly attempts which are all-in-one software packages which we refer to in the manuscript. Many of these packages are admirable and powerful in what they do. However, as illustrated in the table which we have now included (Table 2) and a flowchart (S1 Fig), several of the features we were hoping to use or have access to are not in the present software packages. Further, the main goal of this manuscript was to lay out a replicable and reproducible methodology highlighting using multiple software packages for the sake of flexibility and modularity in several cases and how this was used in our work over several years. We attempted to make this point clearer in the manuscript. As such, this manuscript is to detail the protocol, methodology, and use-cases for a pipeline applied to a large data set and falling under the of goals of “Reproducibility and Replicability in Neuroscience and Mental Health Research”. Further, while portions of this manuscript are mentioned in previous publications, we did not feel we could include the entirety of the approaches as supplemental methods in a purely scientific manuscript without obscuring a large number of details. 

3) The method will be hard to follow; authors make an allusion to Lead-DBS as a comparison – Lead-DBS is a stand-alone Matlab program that does not require any additional downloads or programs and still has a highly active Slack channel for issues; secondly as a guide to the extent that this kind of work should be documented see Stolk et al., 2018 Nature Protocols (which is still not trivial to follow/replicate).

We thank the reviewer for the suggestions. To determine the level of difficulty for following this method and approach, we added two new users who have never dealt with this type of approach before (S3 Fig). Through their feedback, we believe we have improved the protocol as well as demonstrated that this is possible even for individuals who are naïve to the approach. Also, we added a table (Table 2) and a flow chart (S1 Fig) detailing these other software packages for easy comparison between established pipeline. Each of those self-contained software packages have benefits which we take advantage of and attempt to highlight in this methodology. Indeed, the purpose of this protocol is to offer a replicable and reproducible methodology to even naïve users with multiple outputs which is customizable to user needs. We are not trying to make an all-in-one software package, and we have attempted to make this clearer in the manuscript based on your recommendation. We have also incorporated the feedback from our naïve users to improve replicability of the protocol published on protocols.io.

---

## [Decision Letter · Decision Letter 1]

15 Jun 2023

Modular pipeline for reconstruction and localization of implanted intracranial ECoG and sEEG electrodes

PONE-D-22-33193R1

Dear Dr. Soper,

We’re pleased to inform you that your manuscript has been judged scientifically suitable for publication and will be formally accepted for publication once it meets all outstanding technical requirements.

Kind regards,

Federico Giove, PhD

Academic Editor

PLOS ONE

Additional Editor Comments (optional):

Reviewers' comments:

Reviewer's Responses to Questions

**Comments to the Author**

1. Does the manuscript report a protocol which is of utility to the research community and adds value to the published literature?

Reviewer #1: Yes

Reviewer #3: Yes

Reviewer #4: Yes

2. Has the protocol been described in sufficient detail?

To answer this question, please click the link to protocols.io in the Materials and Methods section of the manuscript (if a link has been provided) or consult the step-by-step protocol in the Supporting Information files.

The step-by-step protocol should contain sufficient detail for another researcher to be able to reproduce all experiments and analyses.

Reviewer #1: Yes

Reviewer #3: Yes

Reviewer #4: Yes

3. Does the protocol describe a validated method?

Reviewer #1: Yes

Reviewer #3: Yes

Reviewer #4: Yes

4. If the manuscript contains new data, have the authors made this data fully available?

Reviewer #1: Yes

Reviewer #3: N/A

Reviewer #4: Yes

**5. Is the article presented in an intelligible fashion and written in standard English?**

Reviewer #1: Yes

Reviewer #3: Yes

Reviewer #4: Yes

6. Review Comments to the Author

Reviewer #1: The authors have addressed my concerns. I am particularly pleased with the added validations that were done for the software's usability.

Reviewer #3: The authors have addressed all my concerns and those of the other reviewers very thoroughly and to

my satisfaction, and I recommend publication.

Reviewer #4: Authors have adequately addressed my comments. I do not have any additional comments for the authors.

7. PLOS authors have the option to publish the peer review history of their article (what does this mean?). If published, this will include your full peer review and any attached files.

Reviewer #1: No

Reviewer #3: **Yes: **Abbas Erfanian

Reviewer #4: No

---

## [Editor Report · Acceptance letter]

26 Jun 2023

PONE-D-22-33193R1 

Modular pipeline for reconstruction and localization of implanted intracranial ECoG and sEEG electrodes 

Dear Dr. Soper:

I'm pleased to inform you that your manuscript has been deemed suitable for publication in PLOS ONE. Congratulations! Your manuscript is now with our production department. 

Kind regards, 

on behalf of

Dr. Federico Giove 

Academic Editor

PLOS ONE